# A cortical surface template for human neuroscience

Ma Feilong ●[1]✉, Guo Jiahui ●[1,2], Maria Ida Gobbini ●[3,4] & James V. Haxby ●[1]✉

Neuroimaging data analysis relies on normalization to standard anatomical templates to resolve macroanatomical differences across brains. Existing human cortical surface templates sample locations unevenly because of distortions introduced by inflation of the folded cortex into a standard shape. Here we present the onavg template, which affords uniform sampling of the cortex. We created the onavg template based on openly available high-quality structural scans of 1,031 brains—25 times more than existing cortical templates. We optimized the vertex locations based on cortical anatomy, achieving an even distribution. We observed consistently higher multivariate pattern classification accuracies and representational geometry inter-participant correlations based on onavg than on other templates, and onavg only needs three-quarters as much data to achieve the same performance compared with other templates. The optimized sampling also reduces CPU time across algorithms by 1.3–22.4% due to less variation in the number of vertices in each searchlight.

Various functions of the cerebral cortex are systematically organized on its highly folded surface[1–4]. Functional magnetic resonance imaging (fMRI) data, which were acquired as three-dimensional (3D) volumes, can be projected onto this surface for analysis and visualization in a two-dimensional (2D) space[5]. Compared with the 3D volumetric analysis of fMRI data, surface-based analysis affords better inter-participant alignment, higher statistical power, more accurate localization of functional areas and better brain-based prediction of cognitive and personality traits[6–13]. Due to these advantages, surface-based analysis has been widely adopted by the neuroimaging community, including software[14–17], large-scale datasets[18–20] and cortical atlases and parcellations[21–25].

To account for individual differences in macroanatomy, it is key to normalize all participants' data based on an anatomical template, so that the cortical mesh comprises the same number of vertices across brains, and the same vertex corresponds to the same macroanatomical location. The most commonly used template spaces are fsaverage[5] and fs_LR[26], which were created based on 40 brains. In these standard spaces, the locations of cortical vertices are not based on the anatomical surface, but rather on the spherical surface—a surface obtained by fully inflating each cortical hemisphere to a sphere (Fig. 1c). Then, a geodesic polyhedron—usually a subdivided icosahedron—is used to define the locations of cortical vertices. This procedure allows the vertices to be approximately uniformly distributed on the spherical surface; however, because the geometry of the spherical surface differs from the original surface, the distribution of cortical vertices is far from uniform on the original anatomical surface. For example, cortical vertices are much denser in the central sulcus and the lateral sulcus than in ventral temporal and prefrontal cortices (Fig. 1a and Extended Data Fig. 1).

In this work, we present the onavg (short for OpenNeuro Average) surface template, a human cortical surface template that affords uniform sampling of the cortex. The onavg template was created using high-quality MRI scans of 1,031 participants from 30 OpenNeuro datasets[27]—25 times more participants than previous surface templates[5,26]. We optimized the vertex locations of the onavg template based on the cortical anatomy of the 1,031 participants, so that these vertices were evenly distributed on the anatomical surface instead of on the spherical surface, affording uniform sampling of the cerebral cortex.

In a series of analyses based on an independent naturalistic movie-viewing dataset[28], we demonstrate the advantages that onavg

[1]Center for Cognitive Neuroscience, Dartmouth College, Hanover, NH, USA. [2]School of Behavioral and Brain Sciences, University of Texas at Dallas, Richardson, TX, USA. [3]Department of Medical and Surgical Sciences, University of Bologna, Bologna, Italy. [4]IRCCS Istituto delle Scienze Neurologiche di Bologna, Bologna, Italy. ✉e-mail: feilong.ma@dartmouth.edu; james.v.haxby@dartmouth.edu

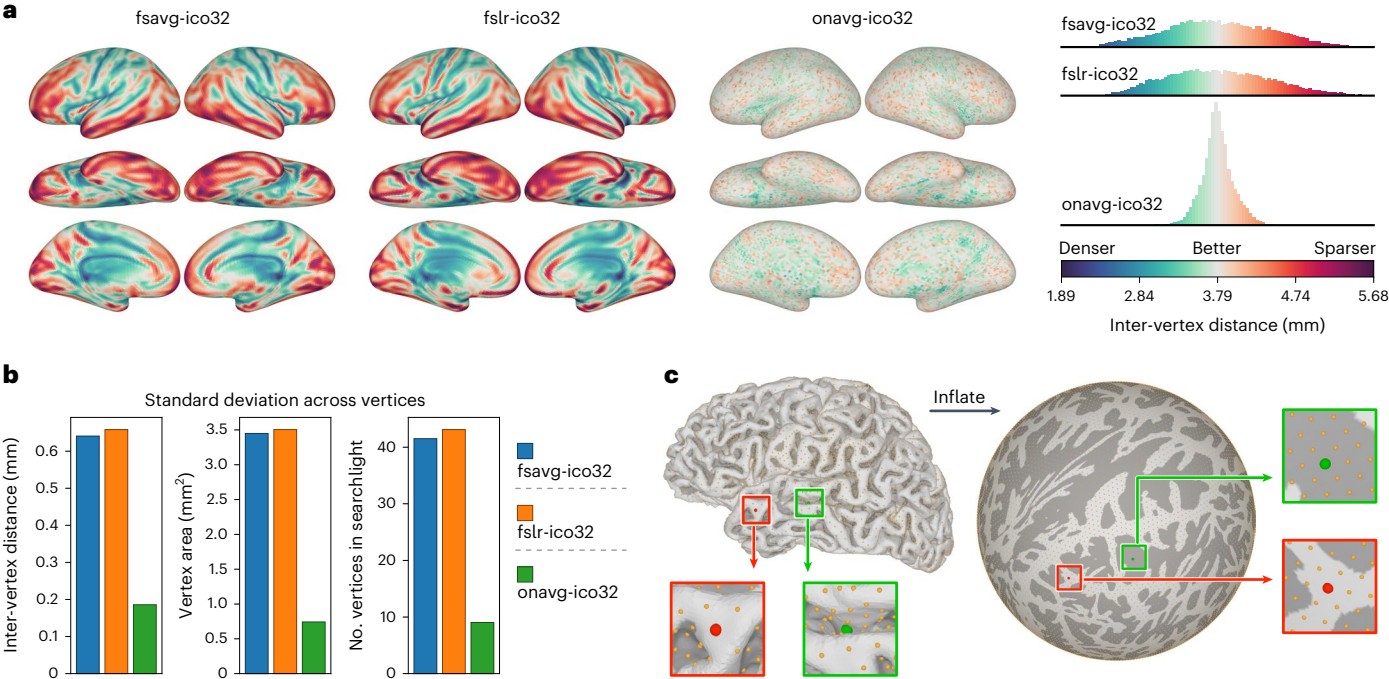

**Fig. 1 | Variation in vertex properties across the cortex. a**, The distribution of vertices in fsavg, fslr and onavg, as measured by inter-vertex distance. **b**, Standard deviation of inter-vertex distance, vertex area and number of vertices in a 20-mm searchlight for fsavg, fslr and onavg. **c**, Classic surface templates sample the cortical surface based on the spherical surface, which was obtained by fully inflating the original anatomical surface. For these templates, the distribution of vertices is almost uniform on the spherical surface (right), but far from uniform on the anatomical surface (left), due to the geometric distortion introduced by inflation. Vertices of the same color (red/green; also in zoomed-in views) are homologous for the two surfaces.

offers using various multivariate pattern analysis (MVPA) techniques[29–31]. On one hand, the anatomy-based sampling of onavg affords better access to the information encoded in spatial response patterns, leading to higher accuracy for multivariate pattern classification[32] and higher inter-participant correlation of representational geometry[33,34]. By switching to onavg, the same classification accuracy and inter-participant correlation can be achieved with three-quarters of the original number of participants (Fig. 2). On the other hand, anatomy-based sampling eliminates large searchlights caused by geometric distortions, leading to consistently reduced computational time for computational algorithms[30,35,36] that rely on searchlight analysis[11,37]. We replicated these analyses using different spatial resolutions, different alignment methods, different numbers of participants and two additional datasets[38], and we observed consistent results across all conditions and datasets (Extended Data Figs. 2–5 and Supplementary Figs. 1–5).

## Results

### Not all vertices were created equal

The sphere-based sampling procedure used by traditional surface templates unavoidably leads to inhomogeneous sampling of the cortical surface, as a result of the geometric distortion when each hemisphere is fully inflated to a sphere. The vertices are approximately uniformly distributed on the spherical surface, which means that cortical regions that were expanded during the inflation will be sampled more densely, and cortical regions that were shrunk during the inflation will be sampled more sparsely.

To quantify the distribution of vertices on the cortical surface, we computed the inter-vertex distance for each cortical vertex, where smaller inter-vertex distance indicates denser sampling in the region, and larger distance indicates sparser sampling. For each vertex, we computed the Dijkstra distance between the vertex and its neighbors for each of the 1,031 participants and averaged across participants

and neighbors. We performed all our analyses using both the ico32 (also known as icoorder5 or 10k, with mean inter-vertex spacing of approximately 4 mm) and ico64 (icoorder6 or 41k, approximately 2 mm) resolutions and observed consistent results. We focus on the ico32 results in the main text and provide the ico64 results in the Supplementary Information.

For both fsaverage and fs_LR (fsavg and fslr for short, respectively, here and thereafter), the inter-vertex distance varied substantially throughout the cortex, and the pattern was similar for both templates (Fig. 1a and Extended Data Fig. 1a). The inter-vertex distance was smaller (denser sampling) in the central sulcus, the postcentral sulcus, the superior temporal sulcus, the lateral sulcus and much of the cingulate cortex and the medial wall; the inter-vertex distance was larger (sparser sampling) in the lateral and medial occipital cortex, the lateral and ventral temporal cortex and the lateral and medial prefrontal cortex (Supplementary Fig. 6). In other words, many brain regions that respond in synchronization across participants[30,35,39–42] and regions that involve high-level cognition[43–47] are not sufficiently sampled based on these traditional surface template spaces.

To resolve these issues caused by traditional templates and sphere-based sampling, we created the onavg surface template using anatomy-based sampling. That is, instead of placing the vertices on the spherical surface based on a geodesic polyhedron, we chose the locations of the vertices based on cortical anatomy: we placed all the vertices on the anatomical surfaces of the 1,031 participants and penalized a pair of vertices if they were too close. After minimizing the distance-based loss function, the vertices were approximately uniformly distributed throughout the cortex.

The anatomy-based sampling of the onavg template reduced the heterogeneity of vertices in many ways. For inter-vertex distance, the variance decreased from 0.41 mm$^2$ (fsavg) and 0.43 mm$^2$ (fslr) to 0.03 mm$^2$ (onavg). For the cortical area occupied by each vertex, the variance decreased from 11.90 mm$^4$ (fsavg) and 12.29 mm$^4$ (fslr) to

0.55 mm$^4$ (onavg). For the number of vertices in a 20-mm searchlight, the variance decreased from 1,723.14 (fsavg) and 1,858.20 (fslr) to 81.58 (onavg). For all these three vertex properties that we assessed, the variance across the cortex decreased substantially from other templates to onavg (mean decrease of 94.23%; range 91.59–95.61%).

## Better cortical sampling improves MVPA results

Multivariate pattern analysis (MVPA) comprises algorithms commonly used in computational neuroscience, such as multivariate pattern classification (MVPC)[29,32] and representational similarity analysis (RSA)[31,33]. MVPA relies on the fact that the spatial response pattern for a certain stimulus or condition is stable across repetitions within the same participant or across participants when their data are functionally hyperaligned[30,36,42]. Therefore, the quality of the spatial patterns formed by cortical vertices is key to successful MVPA.

When resampling neuroimaging data using a traditional sphere-based template, the uneven distribution of vertices on cortical surface creates a systematic bias: brain regions that have smaller inter-vertex distance are densely sampled and overrepresented, and brain regions that have larger inter-vertex distance are sparsely sampled and underrepresented. Note that undersampling a brain region permanently discards certain information, especially the information encoded in fine-grained spatial patterns. Moreover, each vertex has the same weight when computing the pattern vector, and thus the oversampled regions have more influence on the pattern vector compared with the undersampled regions. In other words, the uneven sampling applies an artificial reweighting to cortical regions based on sampling density, which can affect subsequent computational algorithms.

To assess the effects of different surface templates on MVPA, we performed MVPC and RSA on a naturalistic fMRI dataset[28] for each surface space and compared the results. The dataset was collected from 15 participants when they watched the audiovisual movie Forrest Gump in a 3T MRI scanner. We preprocessed the dataset with fMRIPrep[14], which aligns all participants' data based on cortical folding patterns using FreeSurfer. To control for potential confounds from idiosyncratic functional–anatomical correspondence, we repeated the analysis using functionally aligned data based on Procrustes hyperalignment and warp hyperalignment, and we found similar differences between surface templates (Extended Data Figs. 2 and 3 and Supplementary Figs. 1–5). Note that the information loss due to undersampling happened during the resampling step of preprocessing, and thus it affects the results regardless of the alignment method. All analysis was performed using the second half of the movie, independent of the data used for hyperalignment training (first half of the movie).

In the MVPC analysis, we tried to classify which time point of the movie the participant was watching among all 1,781 time points (TRs; 2 s each) based on the whole-brain response patterns. We used a leave-one-participant-out cross-validation and left out a test participant each time. For each time point of the movie, we computed the average response pattern across all other participants as the predicted response pattern of the test participant. Therefore, for each test participant, we had 1,781 measured response patterns and 1,781 predicted response patterns. We examined whether the measured response pattern for a certain time point had the highest correlation to the predicted pattern for the same time point among all 1,781 predicted response patterns (chance accuracy < 0.1%). The average accuracy across participants significantly increased from 13.3% (fsavg) and 13.2% (fslr) to 15.7% (onavg), both $t(14) > 10.0$, Cohen's $d > 2.60$, $P < 10^{-7}$ (paired $t$-tests). For all 15 out of 15 participants, the accuracy based on onavg was higher than based on other templates (Fig. 2a).

The classification accuracy for between-participant MVPC depends on the number of participants. Averaging across a larger number of participants reduces the noise in the predicted response patterns of the test participant, which improves classification accuracy. For all three surface templates, MVPC accuracy consistently increased

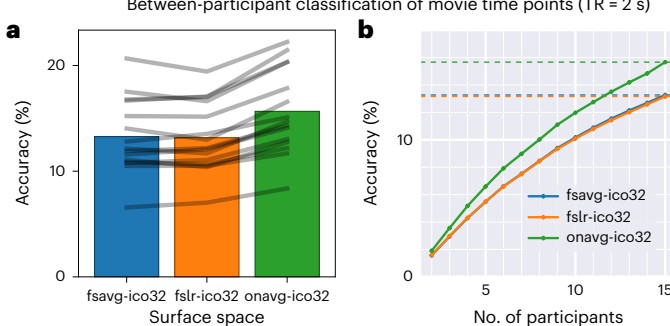

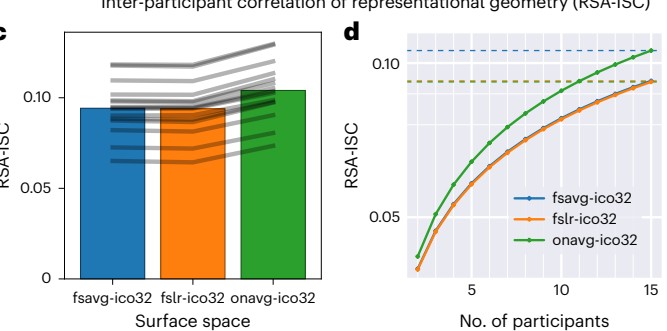

**Fig. 2 | Better cortical sampling improves MVPA results. a**, The between-participant classification accuracy of movie time points based on fsavg, fslr and onavg. Bars denote the average accuracy across all 15 participants and gray lines denote the accuracies of individual participants. **b**, Classification accuracy as a function of the amount of data (the number of participants). Dashed horizontal lines denote accuracies when $n = 15$. **c**, RSA-ISC, computed as the correlation between one participant's RDM and the average of others', based on fsavg, fslr and onavg. Bars denote the average RSA-ISC across 15 participants and gray lines denote those of individual participants. **d**, RSA-ISC as a function of the amount of data. Dashed horizontal lines denote RSA-ISC when $n = 15$.

with more participants (Fig. 2b). Note that the same accuracy for fsavg and fslr with $n = 15$ (13.3% and 13.2%, respectively) is approximately the same as the accuracy for onavg with $n = 11$–$12$ (12.8–13.5%) or $n = 11.7$ and $n = 11.5$, respectively, based on spline interpolation. In other words, the onavg surface template only requires 77.9% and 76.9% of the number of participants for fsavg and fslr, respectively, to achieve the same classification accuracy.

In the RSA analysis, for each searchlight (20 mm), we computed a time-point-by-time-point representational dissimilarity matrix (RDM) for each participant using correlation distance. We computed the Pearson correlation between each participant's RDM and the average of others, which is the inter-participant correlation (ISC) of representational geometry[35] that is often used as the lower-bound of noise ceiling estimation[34]. We refer to this correlation as RSA-ISC here and thereafter. We averaged the RSA-ISCs across all searchlights and obtained an average RSA-ISC for each participant. The average RSA-ISC based on the onavg template was consistently higher than the average RSA-ISCs based on fsavg and fslr for all 15 participants, and the average RSA-ISC significantly increased from 0.094 (fsavg) and 0.094 (fslr) to 0.104 (onavg), both $t(14) > 28.2$, Cohen's $d > 7.30$, $P < 10^{-13}$ (paired $t$-tests; Fig. 2c).

Similar to between-participant MVPC accuracy, the RSA-ISC also benefits from the reduction in noise by averaging over a larger number of participants. For all three surface templates, the RSA-ISC consistently increases with more participants (Fig. 2d). The same RSA-ISC for fsavg and fslr with $n = 15$ (0.094 and 0.094, respectively) is approximately the same as the RSA-ISC for onavg with $n = 11$ (0.094) or $n = 11.0$ and $n = 10.9$, respectively, based on Spearman–Brown interpolation. In

Computational time of representative MVPA algorithms

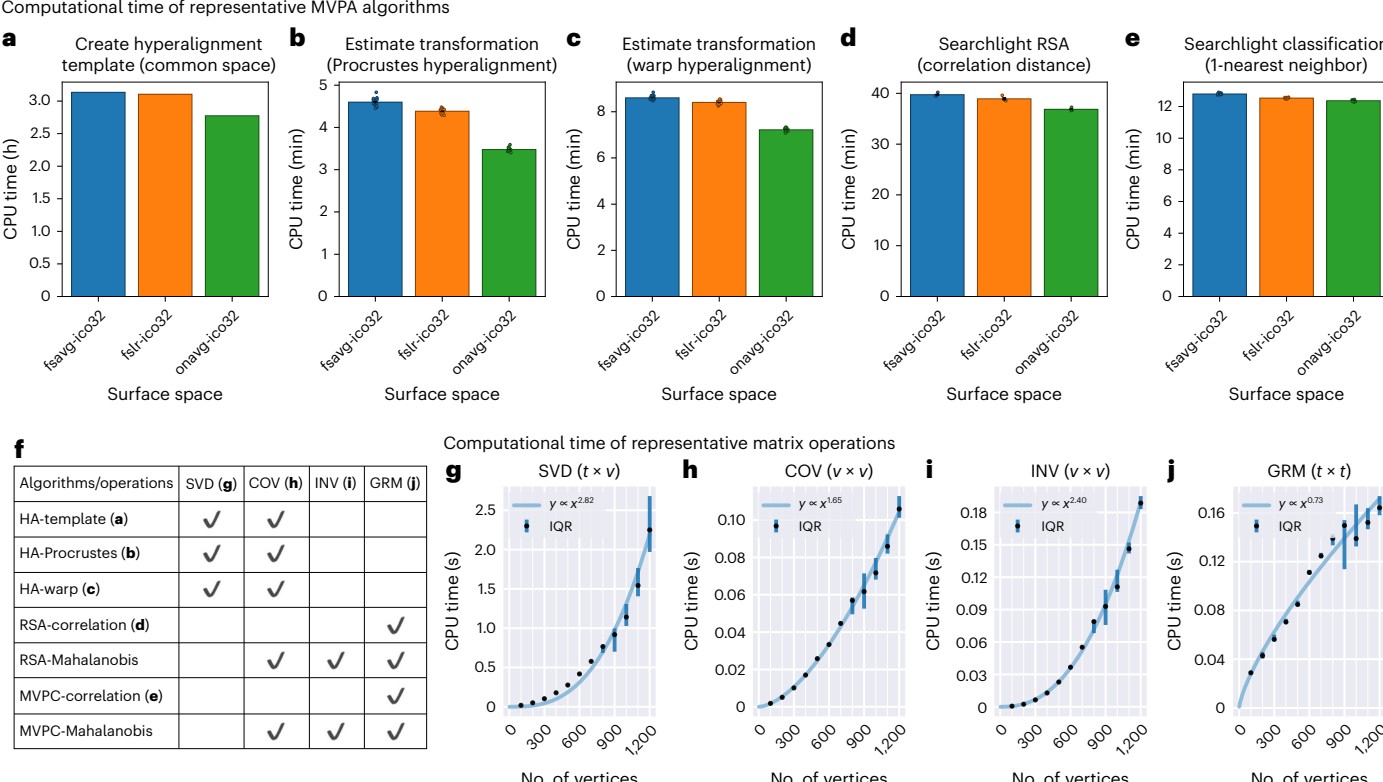

**Fig. 3 | Computational time of representative computational algorithms.**
**a**–**e**, The computational time based on fsavg, fslr, and onavg for various searchlight-based algorithms, including creating hyperalignment common space (**a**), aligning individual participants to the common space (**b**,**c**), RSA (**d**) and multivariate pattern classification (**e**). **f**–**j**, These algorithms rely on basic matrix operations (**f**) and the computational time of these matrix operations is longer when there are more vertices in a searchlight (**g**–**j**). Each light blue curve represents the power function fitted between the number of vertices in a searchlight (the base) and the computational time. The exponent varies between 0.73 and 2.82 for different matrix operations. The vertical blue lines denote the interquartile range of 100,000 repetitions and the central black dots denote the median of the distribution. SVD, singular value decomposition; COV, covariance matrix; INV, inverse of covariance matrix; GRM, Gram matrix. Dot plots represent individuals for **b**,**c**,**e** ($n$ = 15 participants) and repetitions for **d** ($n$ = 3 alignment methods). Error bars denote mean values ± s.e.m. and some error bars are too small to be visible.

other words, the onavg surface template only requires 73.5% and 72.8% of the number of participants for fsavg and fslr, respectively, to achieve the same RSA-ISC.

In both between-participant MVPC and RSA-ISC, improvements in performance were unevenly distributed, with greater improvement in sparsely sampled and inhomogeneously sampled cortical fields (for example, medial occipital, ventral temporal, premotor and insular cortices; Extended Data Fig. 6), indicating that other templates bias the anatomical distribution of results from multivariate pattern analyses.

The improvement of MVPC accuracy and RSA-ISC was consistent across individuals—onavg outperformed fsavg and fslr for all 15 participants (Fig. 2a,c). This was likely because anatomy-based sampling improved every participant's data. We repeated our analysis using different resolutions, different alignment methods, different sample sizes and two additional datasets, and we observed consistent results (Extended Data Figs. 2–5 and Supplementary Figs. 1–5).

In the analyses above, we demonstrate the advantages of the onavg template using MVPA, which by definition relies on spatial patterns. These advantages, in theory, generalize to any neuroscientific data analysis which involves sampling density, uniformity or spatial patterns on the cortical surface. To demonstrate the broad applicability of the onavg template, we used the Human Connectome Project (HCP) dataset to showcase the advantages of the onavg template on three key topics of neuroscience: (1) resting-state functional connectivity, which is commonly used to study the intrinsic functional organization of the brain (Extended Data Fig. 7); (2) functional contrast maps, which

is often used to localize functional regions of interest (Extended Data Fig. 8); and (3) individual differences in brain functional architecture, which is key to precision neuroscience and translational neuroscience (Extended Data Fig. 9). Together, these results demonstrate that the onavg template affords various advantages for a wide range of neuroscientific studies, and these advantages are consistent across datasets and methodological choices.

### Expedited computations for searchlight-based algorithms

Searchlight analysis[11,37] is widely used in combination with MVPC or RSA to assess which part of the brain contains the information of interest, and it serves as the backbone of computational algorithms such as searchlight hyperalignment[35,36,39]. Searchlights are defined as the group of vertices that are within a certain distance (the searchlight radius) from a center, and analyses are computed for overlapping searchlights. Traditional surface templates have large variation in inter-vertex distance across the cortex and, as a result, large variation in the number of vertices in a searchlight (Fig. 1b). The densely sampled brain regions have more vertices in each searchlight, causing prolonged computations in these searchlights.

We systematically assessed the computational time for various searchlight-based algorithms, and we observed a consistent effect that the computational time based on the onavg surface template was shorter than the computational time based on fsavg and fslr, with a 1.3–24.4% reduction in CPU time (Fig. 3; see Extended Data Fig. 10 for results based on ico64 resolution). For creating the common space

for hyperalignment, the CPU time decreased from 3.13 h (fsavg) and 3.10 h (fslr) to 2.77 h (onavg). For hyperalignment of each participant to the common space, the CPU time decreased from 4.60 min (fsavg) and 4.38 min (fslr) to 3.48 min (onavg), based on the classic Procrustes algorithm[42] and from 8.60 min (fsavg) and 8.41 min (fslr) to 7.22 min (onavg), based on the warp hyperalignment algorithm[36]. For searchlight-based RSA analysis, the CPU time decreased from 39.7 min (fsavg) and 38.9 min (fslr) to 36.9 min (onavg). For searchlight-based classification analysis, the CPU time decreased from 12.8 min (fsavg) and 12.5 min (fslr) to 12.4 min (onavg). On average across conditions, switching to onavg led to a 11.5% reduction in CPU time.

The reduction in CPU time is because these computational algorithms rely on matrix operations of the input data matrix (Fig. 3f), and the time required by these matrix operations grows exponentially with the number of vertices (Fig. 3g–j). As a result, a searchlight with an excessive number of vertices will lead to prolonged matrix operations, and eventually, prolonged CPU time for computational algorithms. For example, if the computational time is proportional to the number of vertices squared (quadratic complexity), doubling the number of vertices requires four times as much computational time. For representative matrix operations, the exponent varies between 0.73 and 2.82, and the computations are up to 7.07 times as long when the number of vertices is doubled (Fig. 3g–j). The onavg template avoids making searchlights with an excessive number of vertices created by geometric distortions and uneven sampling (Fig. 1b) and, therefore, it avoids the unnecessary prolonged computations and speeds up computational algorithms substantially.

## Discussion

In this work we introduce the cortical surface template onavg, which was built to achieve uniform sampling of cortical vertices based on high-quality structural scans of 1,031 brains. Compared with classic templates that rely on sphere-based sampling, onavg reduces bias in searchlight analyses, facilitates the efficient use of neuroimaging data and improves the results of various MVPA algorithms. Furthermore, onavg avoids searchlights with an excessive number of vertices caused by uneven sampling and expedites computational methods based on searchlight analysis.

Classic templates create searchlights in densely sampled regions with an excessive number of vertices and searchlights in sparsely sampled regions with too few vertices. Uneven sampling applies an artificial reweighting to cortical regions based on sampling density. For example, ventral and inferior temporal and prefrontal cortices are consistently undersampled in these templates, systematically biasing results of searchlight multivariate analyses in these regions by diminishing power. The more homogeneous searchlight sampling of cortex in the onavg template largely remedies this bias by reducing the variance in number of vertices per searchlight by over 95%.

Replicability and reproducibility are key to neuroscientific research[48–52], and one of the best practices to increase replicability and reproducibility is to use larger amounts of data[53–55]; however, this is often infeasible in practice due to the cost and human effort needed to collect and curate fMRI data. Alternatively, making more efficient use of existing data also increases statistical power, and in turn, better replicability and reproducibility[45,56]. The onavg template provides a way to make better use of neuroimaging data in surface-based analysis. It consistently improved the results of MVPA algorithms across different participants, different data resolutions, different alignment methods, different amounts of data and different datasets. Compared with commonly used surface templates, onavg only requires three-quarters of the amount of data to achieve the same level of performance. Therefore, onavg has the potential to both improve the replicability and reproducibility of future neuroscientific research and reduce the cost and effort of neuroimaging data collection.

Besides improved performance, onavg also reduces the computational time of various MVPA algorithms, which depend on the number of vertices in a searchlight. The geometric distortion of sphere-based sampling creates searchlights in densely sampled regions with an excessive number of vertices, which can be avoided by switching to anatomy-based sampling, which we used to create onavg. Therefore, onavg avoids prolonged computations in large searchlights. Due to the size of movie time point RDMs ($1,781 \times 1,781$), we only benchmarked the RSA computational time based on correlation distance, whose effect size might be smaller than those of alternative distance metrics. For example, the crossnobis distance[57] requires the computation of the covariance matrix and its inverse. These operations have higher time complexity, which could benefit more from avoiding the sizable searchlights created by geometric distortion.

The onavg template was created based on high-quality structural scans of 1,031 brains, more than 25 times more brains as compared with previous surface templates. This was a direct benefit from open science, especially the datasets hosted on OpenNeuro[27] and managed by DataLad[58]. We have made the onavg template openly available under the Creative Commons CC0 license and released it as a DataLad[58] dataset on GitHub (https://github.com/feilong/tpl-onavg). The onavg template is also integrated into TemplateFlow[59] (https://github.com/template-flow/tpl-onavg), the standard repository for neuroimaging templates.

## Online content

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

## Methods

### OpenNeuro datasets

We built the onavg surface template based on high-quality structural scans of 1,031 brains aggregated from 30 OpenNeuro datasets[27]. The datasets and participants were selected based on a few criteria:

1. The participant has at least a high-quality T1-weighted scan and a high-quality T2-weighted scan. That is, the T1-weighted and T2-weighted scans both have (1) whole coverage of the cerebral cortex; (2) a spatial resolution of 1 mm or less in all directions; and (3) no major quality issues.

2. The participant's structural scans show no visible lesion or abnormality. For 'ds002799' only preoperative scans are included.

3. The structural workflow of fMRIPrep successfully finishes within 100 h of CPU time and without errors, and the reconstructed cortical surface has no major artifacts. Longer sessions are predominantly caused by prolonged 'mris_fix_topology' and that usually indicates problematic structural images.

4. The dataset is released under the CC0 license or the Public Domain Dedication and License.

After screening, 1,154 participants passed our criteria. We noted that some of the participants were duplicates. For example, the same individual might have participated in multiple experiments and appear in multiple datasets. To find these duplicates, we compared the similarities of reconstructed cortical surfaces and found 1,031 unique participants. The duplicate participants identified by reconstructed surfaces are consistent with the documentation of the corresponding datasets.

During analysis, we averaged the multiple reconstructed surfaces of the same participant and created an average surface for each participant.

### Participant demographics

One advantage of using 30 OpenNeuro datasets to create the onavg template is that the participants are aggregated through diverse studies, making them representative of the general population of neuroimaging study participants. We did not recruit new participants for this study, and therefore it does not involve informed consent or participant compensation.

Among the 1,031 participants, 471 were female, 408 were male, 2 were nonbinary and 150 were unknown. The mean age ± s.d. was 28.42 ± 14.57 years, based on the 767 participants whose age information was available. The participants were mainly young adults, with a small proportion of younger and older participants. The age range in years was 8–81 and the 10th and 90th percentiles were 18 and 43.76, respectively.

### Preprocessing and surface reconstruction

We downloaded and managed the data files of the 30 OpenNeuro datasets using DataLad[58], and preprocessed the structural scans of these 1,031 participants using fMRIPrep[14] v.21.0.1. Specifically, we used the '--anat-only' option and designated fsaverage as the output space. These settings allowed us to reconstruct the cortical surfaces of these participants using FreeSurfer[15] (build stamp, freesurfer-Linux-centos6_x86_64-stable-pub-v.6.0.1-f53a55a) while benefiting from the optimized structural preprocessing workflow of fMRIPrep. To increase the replicability of our results, we also used the '--skull-strip-fixed-seed' option with a random seed of 0. This ensures that anyone could regenerate identical cortical surfaces as those used in this work.

### Optimize the template using anatomy-based sampling

We optimized the vertex locations of the onavg template, so that no vertices were too close to each other, and the vertices were approximately uniformly distributed throughout the cortex. In the optimization, we used a distance-based objective function, which penalizes pairs of vertices if they were too close. We first performed a coarse discrete optimization based on a geodesic grid, which chose a set of vertex locations that minimized the loss function from a larger set of candidate locations. We then performed a fine optimization, which allowed the vertices to move freely nearby in small steps to further reduce the loss function. We implemented the optimization algorithm in Python using SciPy[60] and NumPy[61].

**Objective function.** We defined the objective function (loss function) of the optimization process as:

$$L = \sum_{\substack{d_{i,j} < d_{\mathrm{thr}} \\ i \neq j}} \frac{1}{d_{i,j}^p}$$

where $d_{i,j}$ is the Dijkstra geodesic distance between a pair of vertices $i$ and $j$. The distance was based on the anatomical surfaces of all 1,031 participants, and therefore the objective of the optimization process was to make the vertices evenly distributed on the anatomical surface instead of the spherical surface. This distance was raised to a power of $p$ to further penalize small distances (neighboring vertices being too close). In practice we used $p = 4$, which worked well, thus we didn't explore other options. Our objective was to ensure the distance was not too small for vertices close to each other, and it is computationally heavy to compute and manage all pairwise distances between vertices. Therefore, we employed a cutoff distance $d_{\mathrm{thr}}$ and only included vertex pairs whose distance was smaller than the cutoff distance $d_{\mathrm{thr}}$. We chose $d_{\mathrm{thr}}$ to be 256 mm/ico, which was 8 mm for ico32 and 4 mm for ico64, approximately twice the average inter-vertex distance.

Occasionally, two vertices might appear at the same location during the optimization process, which makes the distance zero and the loss function ill-defined. To avoid this problem, we added a small number $\varepsilon$ to the distance $d_{i,j}$ in the steps where this problem might happen, and we used $\varepsilon = 0.001$ in practice.

$$L = \sum_{\substack{d_{i,j} < d_{\mathrm{thr}} \\ i \neq j}} \frac{1}{(d_{i,j} + \varepsilon)^p}$$

**Coarse optimization of vertex locations.** The coarse optimization was a discrete optimization, where we chose the vertex locations from a large set of candidate locations, so that the loss function was minimized. The candidate locations were the vertex locations of a high-resolution reference surface. We created the high-resolution reference surface by upsampling the fsaverage spherical surface to a higher resolution. Specifically, we used fsavg-ico256 (655,362 vertices per hemisphere) to optimize onavg-ico32 (10,242 vertices per hemisphere) and fsavg-ico512 (2,621,442 vertices per hemisphere) to optimize onavg-ico64 (40,962 vertices per hemisphere). In other words, the locations of the 10,242 vertices were chosen from the 655,362 candidate locations and the locations of the 40,962 vertices were chosen from the 2,621,442 candidate locations. The number of candidate locations was approximately 64 times the number of vertices.

We initialized the vertex locations by randomly choosing candidate locations without replacement. Then, we tried to find better locations for them. Each candidate location had a loss value based on which vertex locations near it had been occupied, and this value was the same value that would be added to the loss function if the location was occupied by a vertex. For each vertex, we first removed it from its current location and updated the loss value of all candidate locations. We then placed the vertex to the location that had the minimal loss value. We looped through all vertices for up to 100 times and updated their locations accordingly. This process might have stopped early if the local optimum was reached before 100 iterations. The order of vertices was randomized during each iteration.

This coarse optimization process was a greedy algorithm. The local minimum might not be the global minimum, and the results depended

on initialization. Therefore, for each hemisphere and each resolution, we repeated the process for 200 times with different random seeds (and different initializations accordingly). We chose the one that had the smallest loss value for further optimization.

**Fine optimization of vertex locations.** We refined the vertex locations after the coarse optimization to further reduce the loss function. This time, instead of predefined locations, we allowed the vertices to move freely nearby. For each vertex, we used numerical differentiation to find the direction of the gradient, and we moved the vertex along the direction to reduce the loss function. We computed new loss values for different step sizes ranging from $2^{-21}$ to $2^{-10}$ ($4 \times 10^{-7}$ and $1 \times 10^{-3}$) and used the optimal step size multiplied by 0.5 as the final step size to update the vertex location. The factor of 0.5 was because the optimization was performed simultaneously across vertices in parallel, and if the optimal was to reduce the distance between two vertices by 1 mm, each of them should only be moved by 0.5 mm.

It is difficult to compute the Dijkstra distance in this case, because the vertex locations of the new surface do not correspond to vertex locations of the high-resolution reference sphere. Therefore, we approximate the distance based on barycentric interpolation. Each vertex is located on a face of the triangular mesh, and its coordinates $\mathbf{c}_i$ can be represented as a weighted sum of the three vertices of the triangle.

$$\mathbf{c}_i = w_{i,a}\mathbf{c}_a + w_{i,b}\mathbf{c}_b + w_{i,c}\mathbf{c}_c, \text{ where } w_{i,*} \geq 0 \text{ and } w_{i,a} + w_{i,b} + w_{i,c} = 1$$

Similarly, say vertex $j$ locates on a triangle whose vertices were $x$, $y$ and $z$:

$$\mathbf{c}_j = w_{j,x}\mathbf{c}_x + w_{j,y}\mathbf{c}_y + w_{j,z}\mathbf{c}_z, \text{ where } w_{j,*} \geq 0 \text{ and } w_{j,x} + w_{j,y} + w_{j,z} = 1$$

We estimate $d_{i,j}$ as

$$\widehat{d_{i,j}} = \sum w_{i,k}w_{j,l}d_{k,l} \text{ for } k = a, b, c, \text{ and } l = x, y, z$$

This allowed us to compute the distance between a pair of vertices at any locations and further fine-tune the vertex locations without being constrained by the reference sphere.

**Optimization of triangular faces**

After finalizing the optimization of vertex locations, we created an initial surface mesh based on these vertices. Specifically, we created a convex hull based on the vertex locations on the spherical surface, and the simplices of the convex hull were the triangular faces of the initial surface mesh.

We wanted to make each triangular face as similar to an equilateral triangle as possible and therefore we optimized the faces to avoid long edges and elongated triangles. Each pair of neighboring faces forms a quadrilateral $ABCD$. When $AC < BD$, we divide the quadrilateral into two triangles $ABC$ and $ACD$; when $AC > BD$, we divide the quadrilateral into two triangles $ABD$ and $CBD$. Note that the edge lengths were computed based on the anatomical surface of the 1,031 participants, rather than the spherical surface. We repeated this procedure until no further optimization can be performed.

For each triangular face, we also changed the order of its three vertices, $A$, $B$ and $C$, so that the cross product of $AB$ and $BC$ is the same direction as the outward normal of the face. The purpose of the step was to make it easier to compute surface normals and make the generated faces more compatible with those generated by FreeSurfer (https://surfer.nmr.mgh.harvard.edu/fswiki/FreeSurferWiki/SurfaceNormal).

Note that the optimization of triangular faces does not affect the vertex locations, the interpolated data or the analysis results. The purpose of the optimization was simply to make the triangular faces of the surface mesh better describe the geometry of the cortical surface.

**Template evaluation**

**Inter-vertex distance and other vertex properties.** The cortical surface mesh comprises a set of cortical vertices, and the vertices are connected by edges, forming triangular faces. For each vertex, we define its neighbors as the vertices connected to it by an edge. We computed the distance between each vertex and its neighbors and averaged across all 1,031 participants and all neighbors. We used this average distance as the inter-vertex distance of the vertex. Therefore, the inter-vertex distance measures the density of vertices in a local area, where smaller inter-vertex distance indicates denser vertices, and larger inter-vertex distance indicates sparser vertices. To evenly sample the cortex, the inter-vertex distance should have minimal variation across all vertices.

We computed the area of each triangular face and divided it equally among the three vertices of the face. In other words, the area occupied by each vertex was a third of the area of all faces comprising the vertex. Therefore, smaller vertex area indicates denser vertices and larger vertex area indicates sparser vertices. Similar to inter-vertex distance, ideally the variation of vertex area should be as small as possible.

For each vertex, we created a searchlight around it, which was the group of vertices that had a geodesic distance of 20 mm or less from the center vertex. The geodesic distance was computed as the average of all 1,031 participants. The number of vertices in a searchlight varies by brain region—the number is larger for regions with denser vertices and smaller for regions with sparser vertices.

All these three vertex properties (inter-vertex distance, vertex area and number of vertices in a 20-mm searchlight) measures the density of vertices in a local brain area. We expect these properties to have larger variation when the cortex is sampled unevenly, and smaller variation when the cortex is sampled evenly. We computed the s.d. of these properties and compared them across different surface templates, and we found the onavg template had much smaller s.d. compared with other templates (Fig. 1).

**Test dataset for MVPA algorithms.** We used the Forrest dataset[28] to evaluate the surface templates. The dataset was part of the phase 2 data of the studyforrest project (https://www.studyforrest.org/), and it includes fMRI data of 15 participants that were collected while the participants watched the feature movie *Forrest Gump*. We preprocessed the dataset with fMRIPrep[14] and resampled them to different surface spaces. The movie was approximately 2-h long, and during the scan it was divided into eight runs. We used the first half of the movie (the first four runs; 1,818 TRs in total; TR = 2 s) to train hyperalignment models, and the second half of the movie (1,781 TRs) to perform the main analysis. Note that the Forrest dataset was not among the 30 OpenNeuro datasets that we used to create the template and therefore it is completely independent of the template creation process.

We also replicated the analyses with two additional datasets, Raiders ($n = 23$) and Budapest ($n = 21$)[38] (Extended Data Figs. 2 and 3). These two datasets were collected with a different fMRI scanner, different protocols, different movies and different participants from the Forrest dataset[28], which the main MVPA results were based on. With the two new datasets, we observed similar advantages of the onavg template as the results based on the Forrest dataset, demonstrating the robustness of onavg's advantages.

**Hyperalignment template creation.** For each surface template space, we created a hyperalignment template, so that all participants' data could be projected into this common template space. In the common template space, idiosyncrasies in functional–anatomical correspondence are resolved and response patterns can be compared across participants. We followed the procedure described previously[36] to create the template. We first created a local template for each searchlight (20-mm radius), and we made both the representational geometry and the topography of the local template reflective of the central tendency of

the group of participants. We then aggregated the local templates and formed a whole-cortex template. This template creation process made heavy use of principal-component analysis (PCA) and the orthogonal Procrustes algorithm[42], which rely on singular value decomposition (SVD) and the computation of covariance matrices (COV).

**Hyperalignment to template.** For each surface template space, we prepared three sets of data based on different alignment methods: surface alignment (no hyperalignment), Procrustes hyperalignment[35] and warp hyperalignment[36]. We performed all hyperalignment training based on the first half of the movie data and estimated the hyperalignment transformations. We then applied these transformations to the test data (second half of the movie), which was independent of the training data. We report the results based on surface alignment in the main text (Fig. 2), and the results based on Procrustes hyperalignment and warp hyperalignment in Supplementary Figs. 1 and 2, respectively. The classification accuracy and RSA-ISC were both higher for warp hyperalignment than Procrustes hyperalignment and surface alignment, as a result of better alignment across individuals. The differences between surface templates were similar for all three alignment methods, and onavg consistently outperformed other surface templates.

Both Procrustes hyperalignment and warp hyperalignment used in this study are based on searchlight hyperalignment. For each participant, we obtained a local transformation for each searchlight (20-mm radius) and combined these local transformations to form a whole-cortex transformation. The estimation of the transformation made heavy use of ridge regression and the orthogonal Procrustes algorithm[42], which rely on SVD and COV.

**Multivariate pattern classification of movie time points.** For each surface template space and each alignment method, we performed a between-participant multivariate pattern classification of movie time points (TRs) based on the whole brain. We used a leave-one-participant-out cross-validation and a nearest neighbor classifier. We also trained a PCA model based on the first half of the movie (training data) and applied it to the second half of the movie (test data), so that the classification was based on normalized principal components (PCs). The number of PCs was also chosen based on the first half of the movie with a nested cross-validation. The test data comprises 1,781 time points, and therefore each participant had 1,781 measured brain response patterns, one for each time point. Each time, we left out a test participant and computed 1,781 predicted response patterns of the test participant, one for each time point, by averaging the response patterns across other participants. For each measured response pattern, we computed its correlation with all 1,781 predicted response patterns and predicted which time point the participant was watching based on which predicted pattern had the highest correlation. In other words, there were 1,781 choices for this classification task, and the classification was only correct if the corresponding predicted pattern had the highest correlation with the measured pattern. Therefore, the chance level was less than 0.1%. In practice, this classification task can be performed using a correlation-based similarity matrix (1,781 × 1,781), which is a Gram matrix based on the normalized response patterns.

A successful classification relies on the quality of the predicted patterns, and the quality can be improved by averaging over a larger amount of data (averaging over more training participants), which reduces the noise relative to signal. We repeated the classification analysis with smaller numbers of participants, and correspondingly, smaller numbers of training participants. There are multiple ways to choose a subset of participants from the entire set of 15 participants and therefore, for each number of participants, we repeated the sampling procedure for 100 times with different random seeds, and we averaged the results across the 100 repetitions.

**Inter-participant correlation of representational geometry.** Similar to the classification analysis, we repeated the RSA analysis for each surface template space and each alignment method. The RSA analysis was a searchlight analysis. For each searchlight (20-mm radius), we computed a time-point-by-time-point RDM for each participant based on correlation distance. The RDM was based on the test data (second half of the movie) and it comprised 1,781 × 1,781 elements. We computed the inter-participant correlation of representational geometry as the correlation between one participant's RDM and the average of others', which we refer to as RSA-ISC. For each left-out test participant, we averaged the RSA-ISC across all searchlights and obtained a single average correlation. When we averaged across multiple correlation coefficients, we used the Fisher transformation to transform the correlation coefficients to zs, which are approximately normally distributed, and we transformed it back after averaging.

Similar to the classification analysis, the quality of an RDM can be improved by averaging over larger numbers of participants, and the quality can be measured by the reliability of the RDM using Cronbach's $\alpha$ coefficient.

Furthermore, based on the Spearman–Brown prediction formula, we can estimate how this reliability changes with the number of participants used in averaging.

$$r_n = \frac{nr_1}{1 + (n-1)r_1}$$

where $r_n$ is the reliability of the RDM obtained by averaging over $n$ RDMs.

In this formula, $r_1$ can be estimated using $r_n$ and $n$, and in this case $n = 14$ (15 participants in total, one left out). After obtaining $r_1$, we can use it to estimate $r_n$ for different choices of $n$. By combining Cronbach's $\alpha$ coefficient and the Spearman–Brown prediction formula, we estimated the reliability of the average RDM, for different numbers of participants.

The correlation between the average RDM and the left-out participant's RDM is proportional to the square root of the average RDM's reliability. Therefore, by estimating the average RDM's reliability, we can estimate the correlation between the two RDMs for smaller numbers of participants (Fig. 2d).

**Computational time of MVPA algorithms.** For all the MVPA algorithms that we performed, we recorded the CPU time with Python's 'time.process_time_ns' function, which affords nanosecond resolution. In this work, we ran the algorithms in single processes and made sure that the measured CPU time was accurate. In scenarios where recording CPU time is not necessary, it is often better to use parallel computing (for example, with Python's 'joblib' package), which reduces the walltime of these algorithms substantially. For the same algorithm, the CPU time of different surface templates was computed on the same node of Dartmouth's Discovery cluster to eliminate potential confounds from hardware and software differences. We repeated each algorithm for different surface template spaces and recorded the CPU time accordingly.

For each surface template space, we created a hyperalignment template for each hemisphere and recorded the CPU time. We summed over the CPU time across both hemispheres and obtained a total CPU time for each surface template space (Fig. 3a). When we estimated the hyperalignment transformations, we recorded the CPU time for each participant and each hemisphere. Similar to hyperalignment template creation, we computed the sum of the CPU time across both hemispheres and obtained a total CPU time for each participant. We used two different hyperalignment algorithms in our analysis, and therefore we repeated this process for each algorithm (Fig. 3b,c, respectively).

We also performed searchlight classification and searchlight RSA for each template space. The searchlight classification analysis was similar to the whole-cortex classification analysis, except each time the data was from a 20-mm searchlight instead of PCs of the entire cortex and we classified 5-TR segments (10 s each) instead of single TRs (2 s each). We recorded the CPU time for each participant and each

hemisphere, and added together the CPU time of both hemispheres. For the searchlight RSA analysis, we recorded the total CPU time of all participants for each searchlight. This was because estimating the RSA-ISC requires all participants' RDMs, and it is impractical to save these RDMs, and therefore we performed the analysis and recorded the CPU time for each searchlight separately, which does not require saving RDMs to storage. We performed searchlight classification and searchlight RSA for all three alignment methods and averaged the CPU time. This was because for the same surface template, the data matrix shape and the searchlights were the same across different alignment methods, and thus the theoretical computational complexity was the same. By averaging across these repetitions, we further reduce the noise in measured CPU time.

**Computational time of basic matrix operations.** Complex computational algorithms are based on basic matrix operations (Fig. 3f). For example, Procrustes hyperalignment relies on SVD of the COV; correlation-based RSA relies on computing the Gram matrix; RSA with alternative distance metrics, such as the crossnobis distance, requires the inversion of the COV.

The computational time of these matrix operations does not grow linearly with the number of vertices, instead, it takes much longer when the number of vertices is large. To better demonstrate this effect, we systematically evaluated the CPU time of these matrix operations as a function of the number of vertices of the data matrix. We generated random data matrices with different numbers of vertices, ranging from 100 to 1,200 with steps of 100. All these matrices had 1,781 time points, which was the same as the test data. For each number of vertices, we executed these matrix operations 10,000 times each with different random data matrices, which were generated with different random seeds.

To better illustrate the relationship between the CPU time and the number of vertices, we fit an exponential curve $y \propto x^p$, where $y$ is the CPU time, $x$ is the number of vertices and $p$ is the exponent. The exponent $p$ is often between 2–3 (Fig. 3g–j). As a result, if a searchlight contains twice as many vertices compared with the average, the computational time for the searchlight would be 4–8 times as long. When the cortex is unevenly sampled, there are densely sampled regions where the number of vertices is particularly high. Furthermore, there are also more searchlights in these regions, also because the region is densely sampled. As a result, all kinds of searchlight analysis spend prolonged computational time in the densely sampled regions, and when the cortex is evenly sampled, the computational time is consistently reduced (Fig. 3a–e), up to 24.4%.

**Reporting summary**

Further information on research design is available in the Nature Portfolio Reporting Summary linked to this article.

## Data availability

The onavg template is available at TemplateFlow, the standard repository for brain templates, as a DataLad dataset (https://github.com/templateflow/tpl-onavg). Additional group statistics based on the 1,031 participants, such as average maps of sulcal depth, curvature, and vertex area, are available through GIN as a DataLad dataset (https://gin.g-node.org/neuroboros/core). Files of the onavg template are released under the CC0 license. The data of the 1,031 participants that were used to create the onavg template are available through OpenNeuro (https://openneuro.org/) as ds000031, ds000201, ds000221, ds000224, ds000256, ds001233, ds001399, ds001499, ds001597, ds002278, ds002320, ds002330, ds002345, ds002382, ds002634, ds002685, ds002702, ds002737, ds002766, ds002799, ds003242, ds003452, ds003465, ds003499, ds003653, ds003701, ds003745, ds003752, ds003787, and ds003849. The Forrest dataset is available through OpenNeuro as ds000113, and it can also be accessed through the studyforrest website (https://www.studyforrest.org/). The Budapest dataset is available

through OpenNeuro as ds003017. The HCP data are available through ConnectomeDB (https://db.humanconnectome.org/).

## Code availability

The code used to create the onavg template and to perform the benchmarking analyses are available through GitHub (https://feilong.github.io/tpl-onavg/). This GitHub Pages website also contains detailed tutorials on how to use the onavg template and how to transform data between onavg and other templates. The code, along with other information provided on the website, is also archived as a Zenodo repository[62].

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

## Acknowledgements

We thank O. Esteban and Y. O. Halchenko for their help in preparing the TemplateFlow release. This work was supported by National Science Foundation grants 1835200 (M.I.G.) and 1607845 (J.V.H.) and National Institute of Mental Health grant 5R01MH127199 to J.V.H. and M.I.G. Data were provided, in part, by the HCP, WU-Minn Consortium (Principal Investigators, D. Van Essen and K. Ugurbil; 1U54MH091657) funded by the 16 National Institutes of Health (NIH) Institutes and Centers that support the NIH Blueprint for Neuroscience Research; and by the McDonnell Center for Systems Neuroscience at Washington University.

## Author contributions

Conceptualization was the responsibility of M.F., G.J., M.I.G. and J.V.H. Data curation was the responsibility of M.F. Formal analysis was the responsibility of M.F. Funding acquisition was the responsibility of M.I.G. and J.V.H. Methodology was the responsibility of M.F. Resources were the responsibility of M.I.G. and J.V.H. Software was the responsibility of M.F. Supervision was the responsibility of M.I.G. and J.V.H. Visualization was the responsibility of M.F. Writing of the original draft was the responsibility of M.F., G.J., M.I.G. and J.V.H. Review and editing were the responsibility of M.F., G.J., M.I.G. and J.V.H.

## Competing interests

The authors declare no competing interests.

## Additional information

**Extended data** is available for this paper at https://doi.org/10.1038/s41592-024-02346-y.

**Correspondence and requests for materials** should be addressed to Ma Feilong or James V. Haxby.

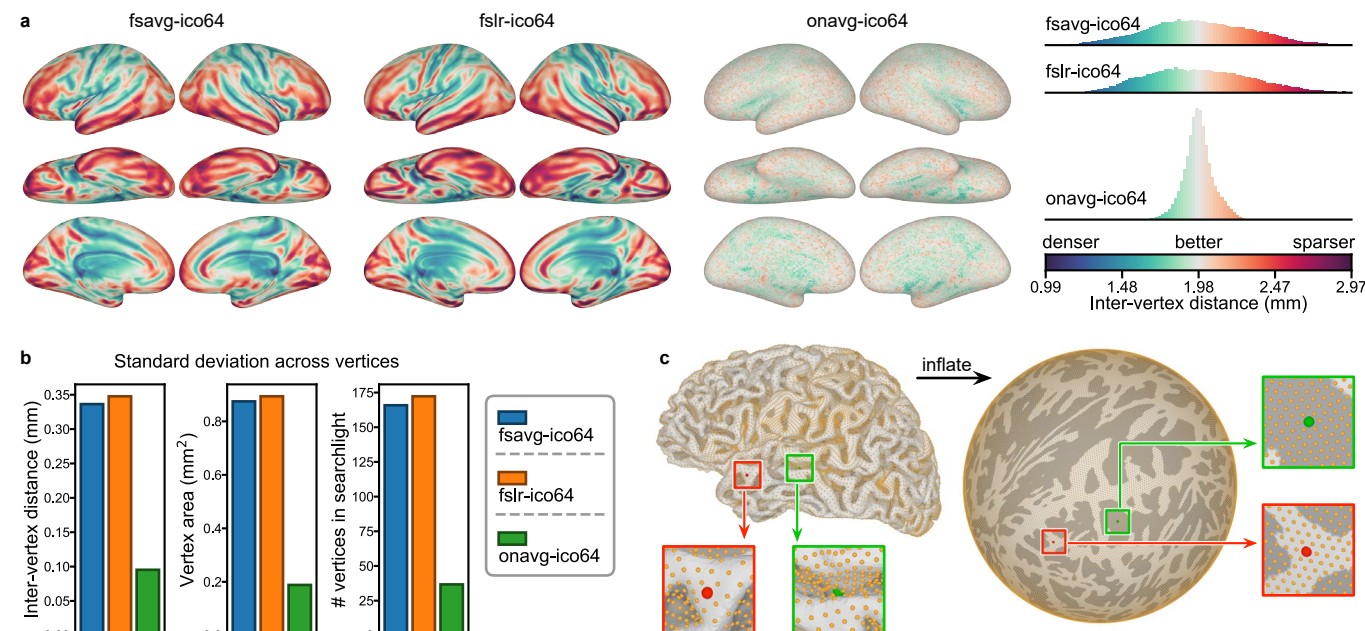

**Extended Data Fig. 1 | Variation in vertex properties across the cortex at ico64 resolution. a**) Distribution of inter-vertex distance on the cortical surface based on fsavg, fslr, and onavg. **b**) The standard deviation of inter-vertex distance, vertex area, and the number of vertices in a 20 mm searchlight based on the three templates. **c**) Example of the distribution of cortical vertices on the anatomical surface and spherical surface based on traditional templates. Vertices of the same color (red/green; also in zoomed-in views) are homologous for the two surfaces.

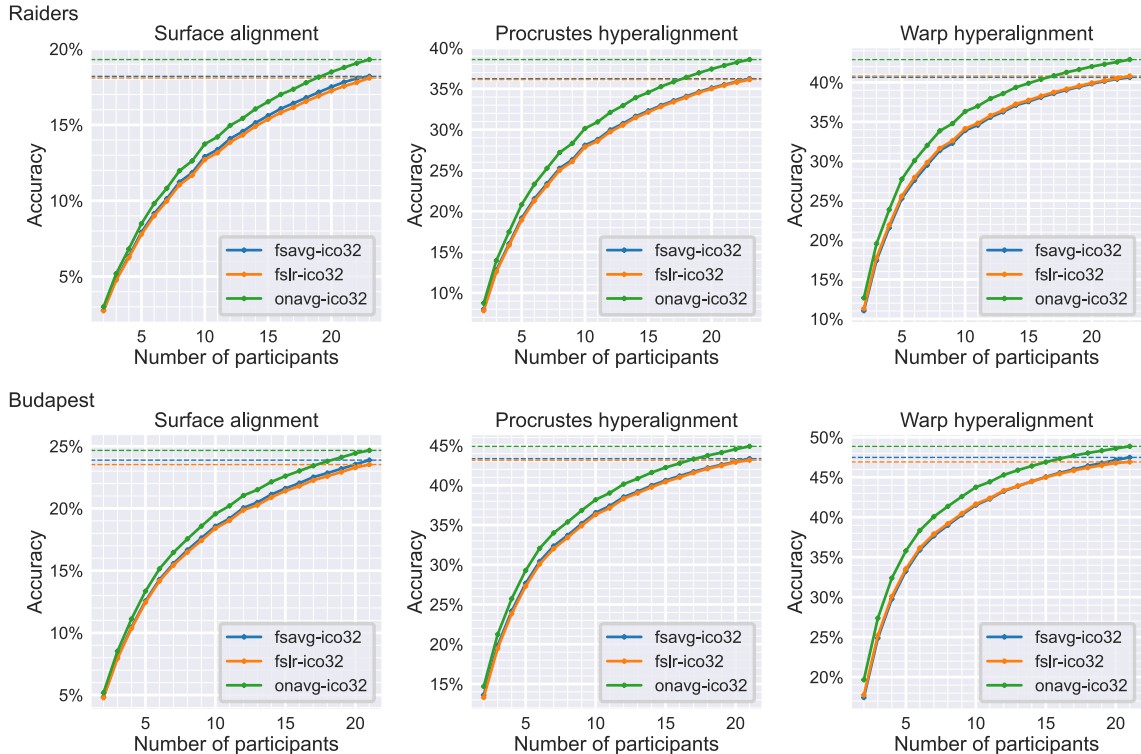

**Extended Data Fig. 2 | Between-participant classification of movie time points.** Classification accuracy as a function of the amount of data (the number of participants) for both the *Raiders* dataset (top) and the *Budapest* dataset (bottom), based on surface alignment (left), Procrustes hyperalignment (middle), and warp hyperalignment (right).

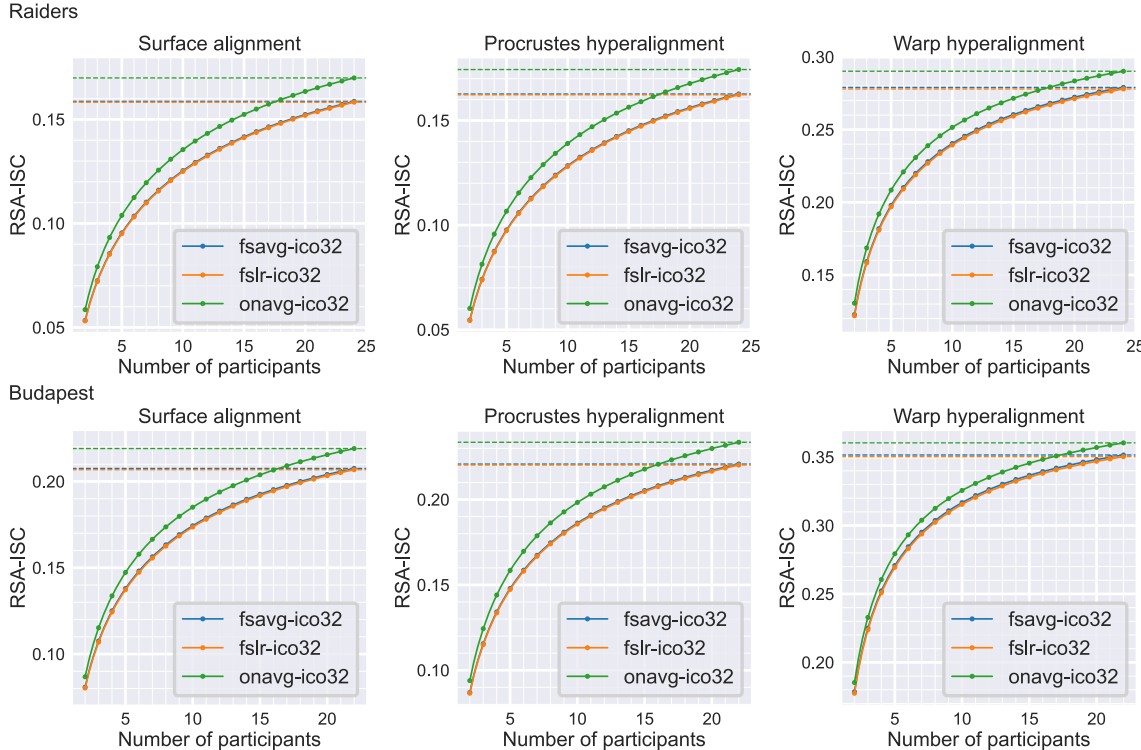

**Extended Data Fig. 3 | Inter-participant correlation of representational geometry (RSA-ISC).** RSA-ISC as a function of the amount of data (the number of participants) for both the *Raiders* dataset (top) and the *Budapest* dataset (bottom), based on surface alignment (left), Procrustes hyperalignment (middle), and warp hyperalignment (right).

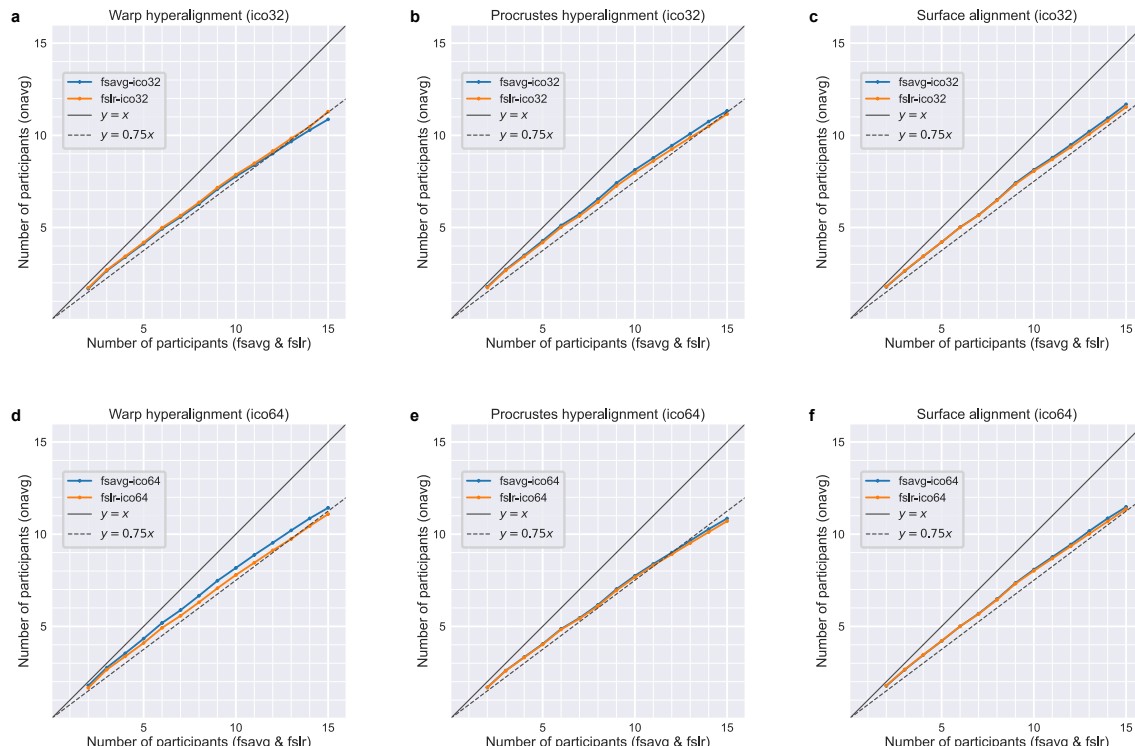

**Extended Data Fig. 4 | The number of participants for onavg (*y*-axis) and other templates (*x*-axis) to achieve the same classification accuracy.** Panels in different rows are based on different data resolutions. Top row: results based on ico32 resolution (**a**–**c**); bottom row: results based on the ico64 resolution (**d**–**f**). Panels in different columns are based on different alignment methods. Left: warp hyperalignment (**a**,**d**); middle: Procrustes hyperalignment (**b**,**e**); right: surface alignment (**c**,**f**).

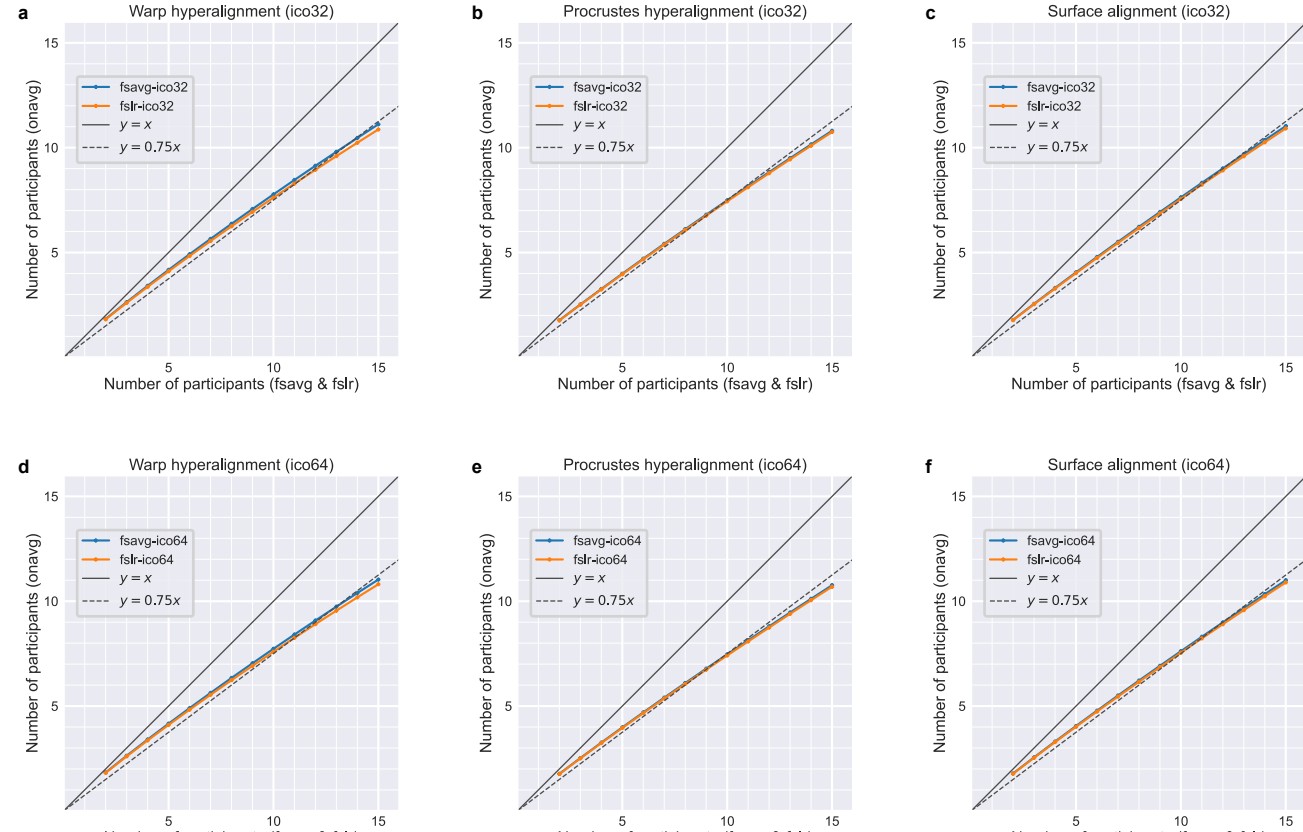

**Extended Data Fig. 5 | The number of participants for onavg (*y*-axis) and other templates (*x*-axis) to achieve the same RSA-ISC.** Panels in different rows are based on different data resolutions. Top row: results based on the ico32 resolution (**a**–**c**); bottom row: results based on the ico64 resolution (**d**–**f**). Panels in different columns are based on different alignment methods. Left: warp hyperalignment (**a**,**d**); middle: Procrustes hyperalignment (**b**,**e**); right: surface alignment (**c**,**f**).

## Between-participant classification of 10-s movie segments

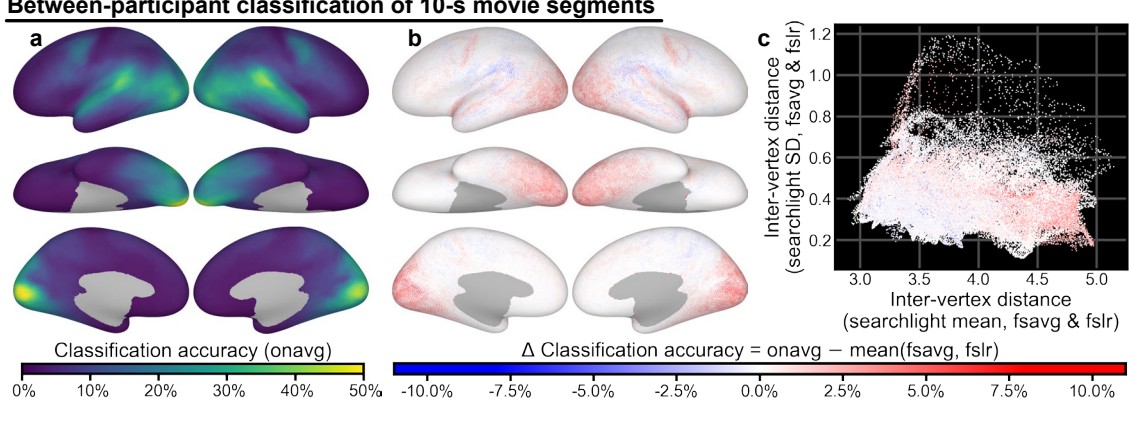

## Inter-participant correlation of representational geometry (RSA-ISC)

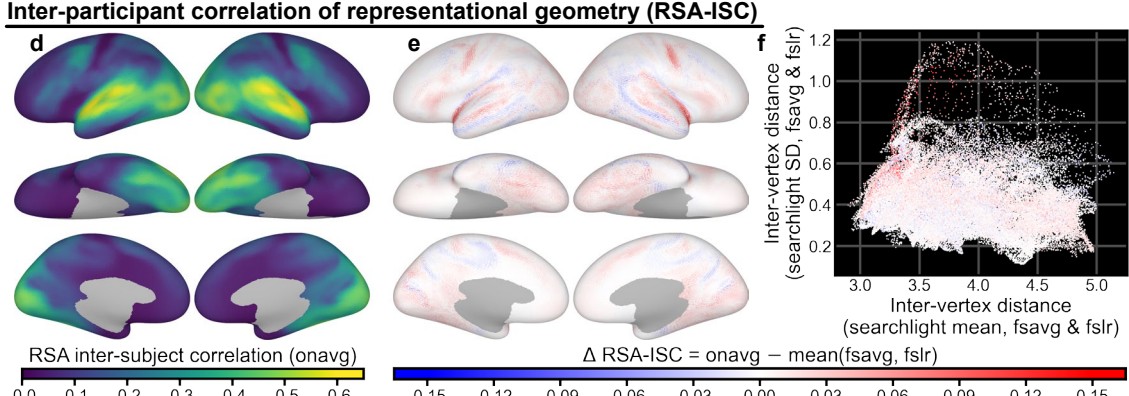

**Extended Data Fig. 6 | Improvement by brain region. a**) Searchlight classification accuracy of 10-s movie segments based on the onavg template. **b**) The difference of classification accuracy between onavg and traditional templates. **c**) The difference of classification accuracy as a function of the regional mean and standard deviation of inter-vertex distance (that is, sampling sparsity and sampling inhomogeneity of the searchlight). **d**) Searchlight RSA-ISC based on the onavg template. **e**) The difference of RSA-ISC between onavg and traditional templates. **f**) The difference of RSA-ISC as a function of the regional mean and standard deviation of inter-vertex distance.

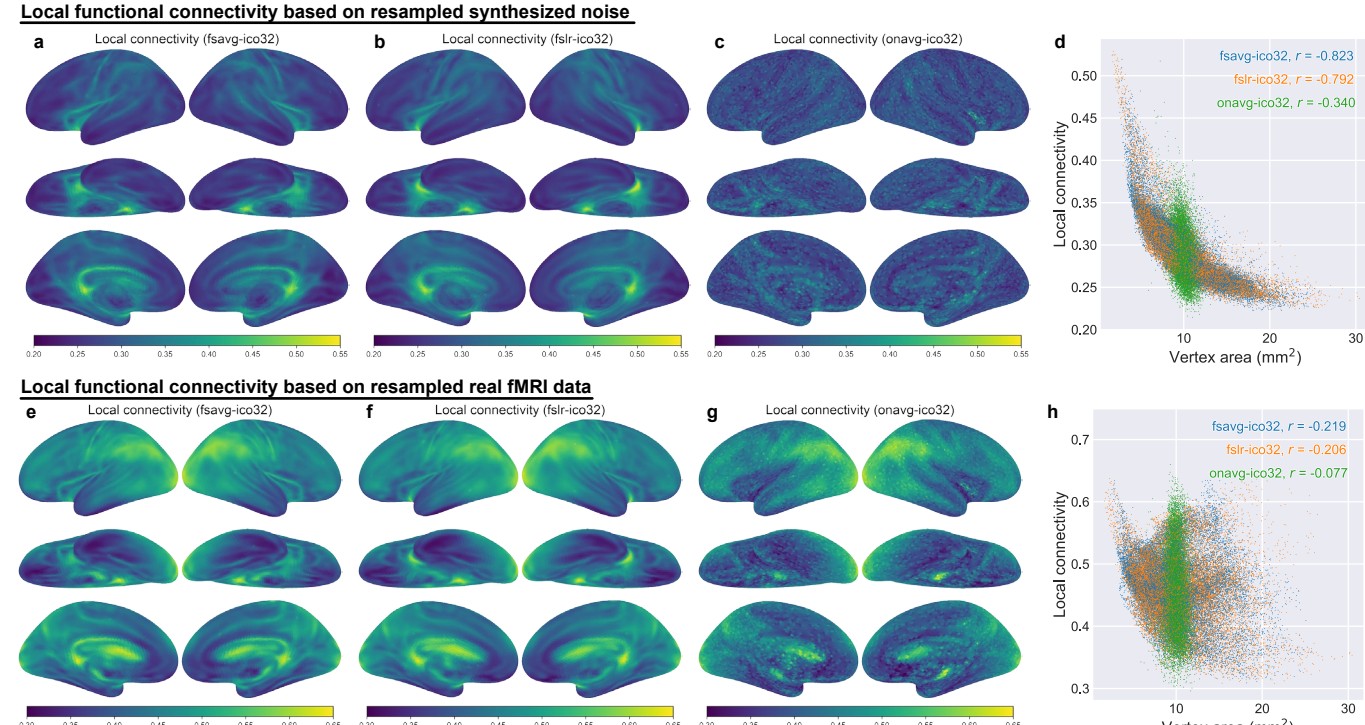

**Local functional connectivity based on resampled synthesized noise**

**a** Local connectivity (fsavg-ico32)  **b** Local connectivity (fslr-ico32)  **c** Local connectivity (onavg-ico32)  **d**

**Local functional connectivity based on resampled real fMRI data**

**e** Local connectivity (fsavg-ico32)  **f** Local connectivity (fslr-ico32)  **g** Local connectivity (onavg-ico32)  **h**

**Extended Data Fig. 7 | Reduced bias in functional connectivity.** When fMRI data are resampled from the initial volumetric acquisition onto the cortical surface, the resampling process creates a systematic bias of local functional connectivity[63]. **a**) Artificial local functional connectivity created by resampling synthesized noise onto cortical surface for fsavg, fslr (**b**), and onavg (**c**). **d**) Local functional connectivity as a function of sampling density (vertex area) based on synthesized noise. **e**) Local functional connectivity based on 3 T resting-state fMRI data of 888 participants from the Human Connectome Project (HCP) for fsavg, fslr (**f**), and onavg (**g**). **h**) Local functional connectivity as a function of sampling density based on resting-state fMRI data.

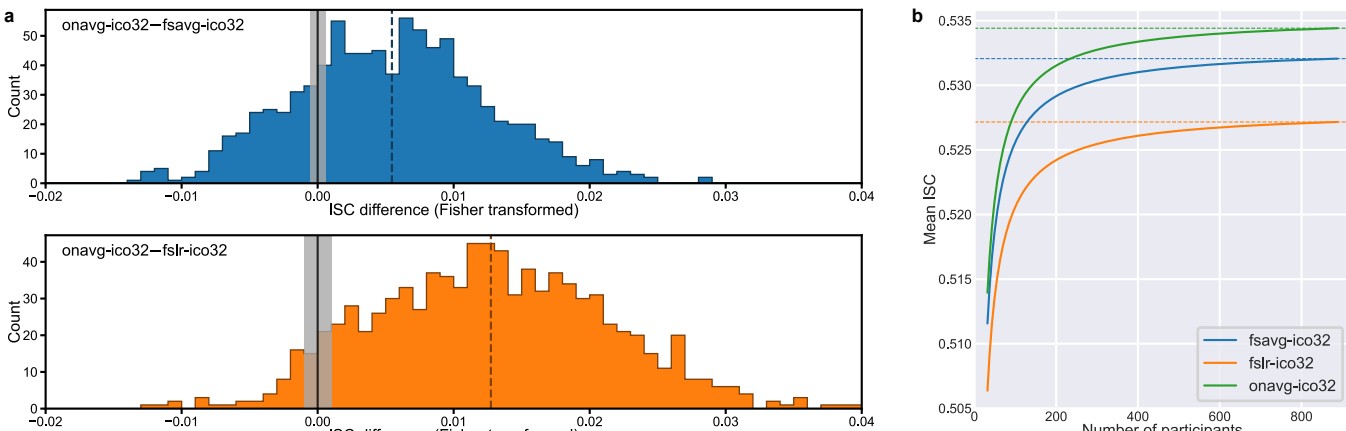

**Extended Data Fig. 8 | Increased ISC of functional contrast maps. a**) The difference of average ISC of functional contrast maps between onavg and fsavg (upper), and between onavg and fslr (lower). The ISC was computed as the correlation between one participant's map and the average of others'. We used the task fMRI contrast maps of the HCP dataset, which includes 47 unique contrasts, covering a wide range of sensorimotor, cognitive, affective, and social regions of the brain. We averaged the correlation coefficients across the 47 contrasts. The dashed line denotes the mean difference ($n$ = 888 participants). The shaded region denotes the center 95% of the null distribution of the mean, based on 100,000 permutations. **b**) ISC of functional contrast maps as a function of the amount of data (the number of participants) for fsavg, fslr, and onavg.

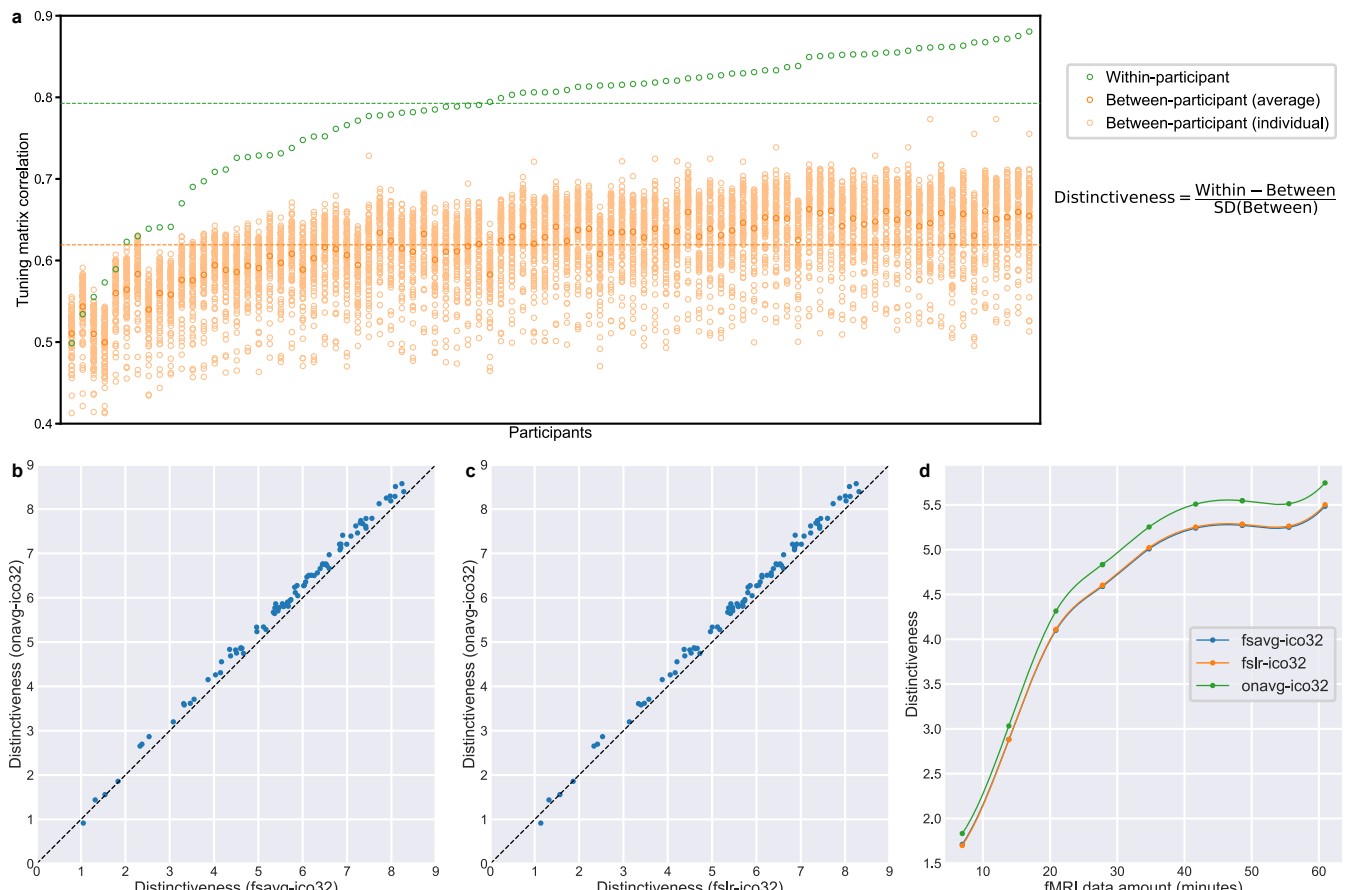

$$\text{Distinctiveness} = \frac{\text{Within} - \text{Between}}{\text{SD(Between)}}$$

**Extended Data Fig. 9 | Amplified individual differences in cortical functional architecture.** For each of the 88 test participants, we obtained two estimates of their neural tuning, one based on each half of the data[36]. **a**) The estimated tuning matrices of the same participant were much more similar than those from different participants. Based on these within-participant and between-participant similarities, we can estimate how distinctive each participant's neural

tuning is. **b**) For almost every participant, the distinctiveness based on the onavg template was higher than those based on fsavg and fslr (**c**) (86 and 84 out of 88, respectively; both $t(87) > 17.8$, $P < 10^{-30}$). **d**) The average distinctiveness across participants as a function of the amount of fMRI data based on fsavg, fslr, and onavg.

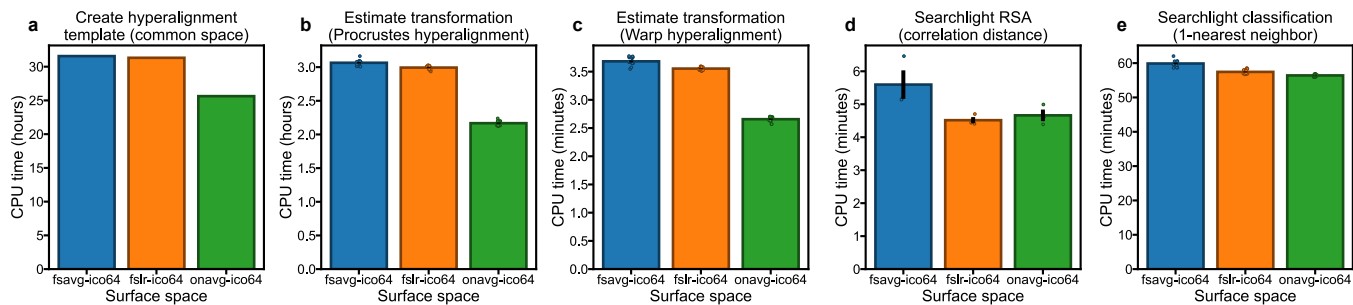

**Extended Data Fig. 10 | Computational time of representative computational algorithms (ico64 resolution).** (**a**–**e**) The anatomy-based uniform sampling of the onavg template affords expedited computations for various searchlight-based algorithms, including creating hyperalignment common space (**a**), aligning individual participants to the common space (**b** and **c**), representational similarity analysis (**d**), and multivariate pattern classification (**e**). Dot plots represent participants for panels **b**, **c**, and **e** ($n$ = 15 participants), and repetitions for panel **d** ($n$ = 3 alignment methods). Error bars denote mean values ± SEM, and some error bars are too small to be visible.

# Reporting Summary

## Statistics

For all statistical analyses, confirm that the following items are present in the figure legend, table legend, main text, or Methods section.

| n/a | Confirmed | |
|---|---|---|
| ☐ | ☒ | The exact sample size (*n*) for each experimental group/condition, given as a discrete number and unit of measurement |
| ☐ | ☒ | A statement on whether measurements were taken from distinct samples or whether the same sample was measured repeatedly |
| ☐ | ☒ | The statistical test(s) used AND whether they are one- or two-sided<br>*Only common tests should be described solely by name; describe more complex techniques in the Methods section.* |
| ☒ | ☐ | A description of all covariates tested |
| ☒ | ☐ | A description of any assumptions or corrections, such as tests of normality and adjustment for multiple comparisons |
| ☐ | ☒ | A full description of the statistical parameters including central tendency (e.g. means) or other basic estimates (e.g. regression coefficient) AND variation (e.g. standard deviation) or associated estimates of uncertainty (e.g. confidence intervals) |
| ☐ | ☒ | For null hypothesis testing, the test statistic (e.g. *F*, *t*, *r*) with confidence intervals, effect sizes, degrees of freedom and *P* value noted<br>*Give P values as exact values whenever suitable.* |
| ☒ | ☐ | For Bayesian analysis, information on the choice of priors and Markov chain Monte Carlo settings |
| ☒ | ☐ | For hierarchical and complex designs, identification of the appropriate level for tests and full reporting of outcomes |
| ☐ | ☒ | Estimates of effect sizes (e.g. Cohen's *d*, Pearson's *r*), indicating how they were calculated |

*Our web collection on statistics for biologists contains articles on many of the points above.*

## Software and code

Policy information about availability of computer code

| Data collection | Data were downloaded from OpenNeuro (https://openneuro.org/) using DataLad 0.17.2 (https://www.datalad.org/). |
|---|---|
| Data analysis | The analysis was performed in a Python 3.8.12 environment with standard scientific computing packages and libraries, including numpy 1.16.4, scipy 1.3.0, pandas 0.24.2, joblib 0.13.2, matplotlib 3.1.0, and seaborn 0.9.0. The data analysis was performed using the Discovery HPC cluster of Dartmouth College. Cortical surface reconstruction was performed using FreeSurfer (freesurfer-Linux-centos6_x86_64-stable-pub-v6.0.1-f53a55a) that was shipped within the fMRIPrep docker image (21.0.1). The code used to create the onavg template and to perform the benchmarking analyses are available through GitHub (https://feilong.github.io/tpl-onavg/). This GitHub Pages website also contains detailed tutorials on how to use the onavg template and how to transform data between onavg and other templates. The code, along with other information provided in the website, is also archived as a Zenodo repository (https://zenodo.org/records/10535655). |

For manuscripts utilizing custom algorithms or software that are central to the research but not yet described in published literature, software must be made available to editors and reviewers. We strongly encourage code deposition in a community repository (e.g. GitHub). See the Nature Portfolio guidelines for submitting code & software for further information.

# Data

Policy information about availability of data

All manuscripts must include a data availability statement. This statement should provide the following information, where applicable:

- Accession codes, unique identifiers, or web links for publicly available datasets
- A description of any restrictions on data availability
- For clinical datasets or third party data, please ensure that the statement adheres to our policy

> The onavg template is available at TemplateFlow, the standard repository for brain templates, as a DataLad dataset (https://github.com/templateflow/tpl-onavg).
> Additional group statistics based on the 1,031 participants, such as average maps of sulcal depth, curvature, and vertex area, are available through GIN as a DataLad dataset (https://gin.g-node.org/neuroboros/core).
> The data of the 1,031 participants that were used to create the onavg template are available through OpenNeuro (https://openneuro.org/) as ds000031, ds000201, ds000221, ds000224, ds000256, ds001233, ds001399, ds001499, ds001597, ds002278, ds002320, ds002330, ds002345, ds002382, ds002634, ds002685, ds002702, ds002737, ds002766, ds002799, ds003242, ds003452, ds003465, ds003499, ds003653, ds003701, ds003745, ds003752, ds003787, and ds003849.  The Forrest dataset is available through OpenNeuro as ds000113, and it can also be accessed through the studyforrest website (https://www.studyforrest.org/).  The Budapest dataset is available through OpenNeuro as ds003017.  The Human Connectome Project data is available through ConnectomeDB (https://db.humanconnectome.org/).

# Human research participants

Policy information about studies involving human research participants and Sex and Gender in Research.

| | |
|---|---|
| Reporting on sex and gender | Out of the 30 OpenNeuro datasets used to create the onavg cortical surface template, 25 reported demographics information. Two datasets (ds000221 and ds000224) reported gender; one dataset (ds001597) did not report sex or gender; the remaining 22 datasets (ds000201, ds000256, ds001233, ds001399, ds001499, ds002278, ds002330, ds002345, ds002382, ds002685, ds002702, ds002737, ds002799, ds003242, ds003452, ds003465, ds003499, ds003653, ds003701, ds003745, ds003787, ds003849) reported sex. For each spatial resolution, we created a single surface template across sex and gender. |
| Population characteristics | Participants were primarily young adults (mean age = 28.42 years, std = 14.57 years). 471 were female, 408 were male, and 2 were non-binary. Out of the 1,031 participants, age was unknown for 264, and sex/gender was unknown for 150. Participants had no visible lesion or structural abnormality. |
| Recruitment | Participants were recruited by the original researchers of the 30 OpenNeuro datasets. |
| Ethics oversight | This study re-analyzes openly available datasets. These datasets were respectively collected by other researchers and approved by their institutions. Details on each dataset can be found on OpenNeuro (e.g., https://openneuro.org/datasets/ds000031), and additional information is available in the "References and Links" section of each dataset. |

Note that full information on the approval of the study protocol must also be provided in the manuscript.

# Field-specific reporting

Please select the one below that is the best fit for your research. If you are not sure, read the appropriate sections before making your selection.

☐ Life sciences  ☒ Behavioural & social sciences  ☐ Ecological, evolutionary & environmental sciences

For a reference copy of the document with all sections, see nature.com/documents/nr-reporting-summary-flat.pdf

# Behavioural & social sciences study design

All studies must disclose on these points even when the disclosure is negative.

| | |
|---|---|
| Study description | We used high-quality structural scans of 1,031 participants to create a cortical surface template that evenly samples the cerebral cortex. |
| Research sample | To maximize the quality of the template, we used as much openly available high-quality data as possible when we started the project in Feb 2022. Specifically, we used OpenNeuro datasets that have a CC0 or PDDL license, and ensured that: (a) each participant has at least one T1w scan and one T2w scan, (b) the scans cover the entire cerebral cortex; (c) the spatial resolution is 1 mm or less in all directions, and (d) the data have no major quality issues. The sample is representative of the population of all neuroimaging research participants, and the demographics of the sample match those of other datasets (see Supplementary Note). The rationale for choosing this sample is that both larger sample size and better data quality will increase the quality of the final template. Therefore, we tried to make the sample size as large as possible while making sure that we only used data that had passed the quality check. In other words, the procedure was designed to maximizing the template quality by achieving a balance between sample size and data quality. |

| | |
|---|---|
| Sampling strategy | The OpenNeuro datasets afford convenience sampling of all participants of neuroimaging research. No sample size calculation was performed, and we aimed to use as much data that pass the quality check as possible. We ended up with a sample size of 1,031 participants, which is 25 times more than previous samples used to build cortical surface templates. Therefore, it is reasonable to believe that the sample size is sufficient for the current study. |
| Data collection | No new data was collected for the current study. The 30 OpenNeuro datasets were collected by their original researchers, and therefore these researchers were blinded to the hypothesis of the current study. The structural scans of these datasets were acquired using 3 T MRI scanners. |
| Timing | Not applicable because we did not collect new data or recruit participants for this study. Not all OpenNeuro datasets have clearly documented start and stop dates of data collection. However, it is unlikely for the structural scans to be affected by the timing of data collection. |
| Data exclusions | We excluded participants whose cortical reconstruction could not be properly performed. This is usually caused by quality issues of the structural scans and accompanied by a prolonged "mris_fix_topology" step of FreeSurfer. |
| Non-participation | Not applicable because we did not collect new data or recruit participants for this study. |
| Randomization | We used leave-one-participant-out cross-validation for the MVPA analyses described below, which does not involve randomization. |

# Reporting for specific materials, systems and methods

We require information from authors about some types of materials, experimental systems and methods used in many studies. Here, indicate whether each material, system or method listed is relevant to your study. If you are not sure if a list item applies to your research, read the appropriate section before selecting a response.

## Materials & experimental systems

| n/a | Involved in the study |
|---|---|
| ☒ | ☐ Antibodies |
| ☒ | ☐ Eukaryotic cell lines |
| ☒ | ☐ Palaeontology and archaeology |
| ☒ | ☐ Animals and other organisms |
| ☒ | ☐ Clinical data |
| ☒ | ☐ Dual use research of concern |

## Methods

| n/a | Involved in the study |
|---|---|
| ☒ | ☐ ChIP-seq |
| ☒ | ☐ Flow cytometry |
| ☐ | ☒ MRI-based neuroimaging |

## Magnetic resonance imaging

### Experimental design

| | |
|---|---|
| Design type | We used the naturalistic movie viewing data of the StudyForrest dataset.<br>In Supplementary Information, we also used the Raiders dataset, Budapest dataset, and the Human Connectome Project (HCP) dataset. |
| Design specifications | Participants watched the feature movie Forrest Gump in the scanner. The movie was divided into 8 parts, and the length of the runs were adjusted accordingly. See Hanke et al. (2016, 10.1038/sdata.2016.92) for details.<br>The two other movie datasets (Raiders, Budapest) had similar designs.<br>The HCP data we used including 3 T resting-state fMRI data, 3 T task fMRI data, and 7 T movie-watching fMRI data. The data are described in detail in its documentation website (https://www.humanconnectome.org/study/hcp-young-adult). |
| Behavioral performance measures | Ratings related to movie watching was collected in the original dataset (Hanke et al., 2016), which was irrelevant for the current study. The HCP dataset has various behavioral measures, which were also irrelevant for the current study. |

### Acquisition

| | |
|---|---|
| Imaging type(s) | T1w and T2w structural scans. Functional scans for the StudyForrest dataset, the Raiders and Budapest datasets, and the HCP dataset. |
| Field strength | 3 T (most data) and 7 T (HCP movie). |
| Sequence & imaging parameters | All structural scans used in the analysis have a spatial resolution of 1 mm or less in all directions. Detailed imaging parameters varied by dataset. |
| Area of acquisition | The structural and functional scans covered the entire cerebral cortex. |
| Diffusion MRI | ☐ Used  ☒ Not used |

## Preprocessing

**Preprocessing software**
We preprocessed all structural data using fMRIPrep 21.0.1 with the `--anat-only` option. The StudyForrest data were preprocessed with fMRIPrep 21.0.2. The Raiders and Budapest datasets were preprocessed with fMRIPrep 20.2.7. We used the minimally preprocessed data of the HCP dataset and removed the noise (See "Noise and artifact removal" below).

**Normalization**
Surface-based normalization was used for all participants.

**Normalization template**
Normalization was performed based on FreeSurfer's group statistics of folding patterns to derive "sphere.reg" for each participant. In the evaluation analysis, functional data were resampled to 3 different surface spaces for each resolution: fsaverage, fs_LR, and the newly created onavg.

**Noise and artifact removal**
A standard set of nuisance variables were regressed out from functional data, including: 6 motion parameters and their derivatives, 6 aCompCor components from WM and CSF, framewise displacement, and polynomial trends up to the 2nd order. For the HCP dataset, we used the average WM and CSF signals instead of the aCompCor components.

**Volume censoring**
The data used all have great quality, and we did not consider volume censoring.

## Statistical modeling & inference

**Model type and settings**
We used a leave-one-participant-out cross-validation scheme for both the between-subject MVPC analysis and the RSA analysis. For the MVPC analysis, we predicted the left out test participant's brain responses based on the other 14 participants, and we examined whether the measured and the predicted response patterns for the same time point (TR) had the highest correlation. For the RSA analysis, for each searchlight, we computed the correlation between the test participant's RDM and the average RDM of other participants. This is the inter-subject correlation (ISC) of the RDMs, which we referred to as RSA-ISC.

**Effect(s) tested**
Our study compared MVPA performance based on different templates, specifically, between onavg and fsavg, and between onavg and fslr. We compared the accuracy for multivariate pattern classification and the inter-subject correlation of representational geometry. We observed that the onavg template outperformed other surface templates in all 15 participants, corresponding to a P-value of 3e-5 based on binomial testing. We reported the statistics for parametric testing in the main text.

**Specify type of analysis:** ☐ Whole brain   ☐ ROI-based   ☒ Both

**Anatomical location(s)**
The MVPC analysis was performed using data from the entire cerebral cortex. The RSA analysis was a searchlight analysis with a searchlight radius of 20 mm.

**Statistic type for inference**
(See Eklund et al. 2016)
The statistics were either classification accuracy based on the entire cerebral cortex, or average RSA-ISC across all searchlights.

**Correction**
This study does not involve multiple comparisons across brain regions.

## Models & analysis

| n/a | Involved in the study |
|-----|------------------------|
| ☐ ☒ | Functional and/or effective connectivity |
| ☒ ☐ | Graph analysis |
| ☐ ☒ | Multivariate modeling or predictive analysis |

**Functional and/or effective connectivity**
Functional connectivity used in the Supplementary Information (HCP 3 T resting-state) was based on Pearson correlation.

**Multivariate modeling and predictive analysis**
See "Model type and settings" section above.

