## [Peer Review File · Nature Methods]

Peer Review Information

Manuscript Title: A cortical surface template for human neuroscience

Corresponding author name(s): Ma Feilong

Editorial Notes: None

Reviewer Comments & Decisions:

Decision Letter, initial version:

Dear Dr Feilong,

Let me first sincerely apologize for the long review process. Your Article, "A cortical surface template for human neuroscience", has now been seen by two reviewers. As you will see from their comments below, although the reviewers find your work of considerable potential interest, they have raised a number of concerns. We are interested in the possibility of publishing your paper in Nature Methods, but would like to consider your response to these concerns before we reach a final decision on publication.

We therefore invite you to revise your manuscript to address these concerns. Specifically, please make sure to show that the template can be used in a variety of different contexts. Please also address the other concerns of the reviewers.

- * include a point-by-point response to the reviewers and to any editorial suggestions
- * please underline/highlight any additions to the text or areas with other significant changes to facilitate review of the revised manuscript
- * address the points listed described below to conform to our open science requirements
- * ensure it complies with our general format requirements as set out in our guide to authors at www.nature.com/naturemethods

* resubmit all the necessary files electronically by using the link below to access your home page

[Redacted]

We hope to receive your revised paper within 4-8 weeks. If you cannot send it within this time, please let us know. In this event, we will still be happy to reconsider your paper at a later date so long as nothing similar has been accepted for publication at Nature Methods or published elsewhere.

OPEN SCIENCE REQUIREMENTS

REPORTING SUMMARY AND EDITORIAL POLICY CHECKLISTS

DATA AVAILABILITY

We strongly encourage you to deposit all new data associated with the paper in a persistent repository where they can be freely and enduringly accessed. We recommend submitting the data to discipline-specific and community-recognized repositories; a list of repositories is provided here:

<http://www.nature.com/sdata/policies/repositories>

All novel DNA and RNA sequencing data, protein sequences, genetic polymorphisms, linked genotype and phenotype data, gene expression data, macromolecular structures, and proteomics data must be deposited in a publicly accessible database, and accession codes and associated hyperlinks must be provided in the “Data Availability” section.

Please include a “Data availability” subsection in the Online Methods. This section should inform readers about the availability of the data used to support the conclusions of your study, including accession codes to public repositories, references to source data that may be published alongside the paper, unique identifiers such as URLs to data repository entries, or data set DOIs, and any other statement about data availability. At a minimum, you should include the following statement: “The data that support the findings of this study are available from the corresponding author upon request”, describing which data is available upon request and mentioning any restrictions on availability. If DOIs are provided, please include these in the Reference list (authors, title, publisher (repository name), identifier, year). For more guidance on how to write this section please see: <http://www.nature.com/authors/policies/data/data-availability-statements-data-citations.pdf>

CODE AVAILABILITY

Please include a “Code Availability” subsection in the Online Methods which details how your custom code is made available. Only in rare cases (where code is not central to the main conclusions of the paper) is the statement “available upon request” allowed (and reasons should be specified).

For more information on our code sharing policy and requirements, please see: <https://www.nature.com/nature-research/editorial-policies/reporting-standards#availability-of-computer-code>

MATERIALS AVAILABILITY

As a condition of publication in Nature Methods, authors are required to make unique materials

promptly available to others without undue qualifications.

ORCID

Best regards,
Nina

Nina Vogt, PhD
Senior Editor
Nature Methods

Reviewers' Comments:

Reviewer #1:

Remarks to the Author:

This paper introduced a new surface template constructed from a large scale of MRI scans consisting of 1000+ subjects. The key difference between this template and other commonly used surfaces, such as fsaverage (FreeSurfer) or fs_LR (HCP), is the uniform sampling of the vertices on the anatomical surface manifold, rather than uniform sampling on the inflated spherical space.

The authors demonstrated the advantage of their surface template via 1) a multivariate pattern classification (MVPC) and 2) a representational similarity analysis (RSA), both on a naturalistic (movie) fMRI dataset (n=15). The results showed a consistent improvement of the accuracy in MVPC and intersubject correlation in RSA, compared to fsavg and fsLr. Furthermore,

the authors also demonstrated that the computational time for the searchlight-based algorithms was reduced using the uniform template.

The idea of uniform sampling on the pial surface rather than the sphere is neat. The coarse-to-fine optimization procedure for the template construction is clear and straightforward. I do think there could be a huge potential in using this onavg template in a variety of neuroscientific tasks.

However, I'm not fully convinced by the demonstrations for the following reasons:

- The experiments should not be constrained by the specific experimental pipelines from authors' previous works. It will be much more convincing if the authors can show the onavg template is able to yield better performance in a wide spectrum of neuroscientific tasks, in order to claim onavg is a better template for "human neuroscience" as their title suggested as well as be suitable to the broad audience in this journal.

- Another downside of using the authors' current experiments is that the experimental pipeline seems too complicated. How could the authors prove it's the onavg template that results in improved performance, rather than some inherent bias of the experimental pipeline?

- The sample size (in the demonstration experiments, not the template construction) is too small.

- The usage of movie fMRI data is kind of tricky, as the brain response can be very complex. I don't see (and it's unlikely based on the boxplots) there will be statistical significance in MVPC accuracy nor in RSA-ISC between onavg result and fsavg/fslr result.

- There are many other domains that also rely on surface template besides naturalistic fMRI studies, such as anatomical/functional parcellations, cytoarchitecture studies linking ex vivo scans to in vivo MRI, vascular studies, etc.

- With all being said above, I would suggest the authors conduct some very very basic experiments that people use every day across several different categories of neuroscientific tasks to demonstrate the benefit of using onavg. Just a few examples I can think of:

- * Registration: register each subject from a large-scale dataset (e.g. HCP or UK Biobank, as long as it's independent of the OpenNeuro set) to the onavg template based on geometric features such as sulcal depth or curvature using any widely used registration tool e.g. FreeSurfer, FSL, ANT, whatever. Once the pipeline is fixed, only change the registration target, i.e. swapping the onavg with fsavg and fslr, and compare the results. Do you see improved registration accuracy?

- * resting-state fMRI: sample (perhaps the minimally preprocessed) rsfMRI data for each subject onto the three templates, simply compute connectivity (vertex-wise Pearson correlation matrix). Do you see a smaller variance in the connectivity across subjects?

- * task fMRI: sample task data onto the three templates and just do a GLM regression against the block designs. Maybe just need one or a few simple tasks for the demonstration, e.g. finger tapping task or visual task where the responsive regions are well localized. Do you see a stronger task response averaged over subjects or do you see sharper boundaries between task and non-task regions?

Some other minor comments:

- The authors claimed, "Note that oversampling a brain region does not provide extra information-the extra vertices are simply extra interpolations from the same acquired data". This seems to be a very bold claim and perhaps only applicable to the current scale of fMRI data. There are many other applications that need denser samples, e.g. studies of high res ex vivo data. That's why when people release data, surfaces are usually provided in different resolutions.

- Results in ico64 (~2mm inter-vertex distance) were shown in supplementary materials. But this resolution is usually considered downsampled surfaces. Does onavg also do better than fsavg/fslr in higher resolution surfaces, e.g. 164k native mesh from FreeSurfer/HCP?

- It would be great to show, in the very first figure, how does the onavg template look like in both the pial and the spherical surface, and perhaps a zoomed-in version and compare that with fsavg/fslr, just settle an intuitive impression for the audience.

Albeit these comments, I find the study is interesting, worth pursuing, and can be a great contribution to the community if the authors can demonstrate its advantage over a range of most commonly used neuroscientific tasks.

Reviewer #2:

Remarks to the Author:

Feilong et al. proposed a new surface template that solves the problem of the current standard surface templates: the vertices are unevenly distributed in anatomical space. With this new template, they showed that the classification accuracy of movie fMRI time points was higher, the inter-subject correlation of representational geometry was higher, and the computational time of searchlight-based algorithms was lower, compared to using standard surface templates. This paper is very well written and the authors demonstrated the applicability of this new template. However, I have a few comments:

1. All evaluation measures involved using hyperalignment. However, many studies do not use hyperalignment. For example, would the movie time points classification accuracy still increase?

2. Both the classification accuracy of movie time points, and the inter-subject correlation of representational geometry look for the common neuro-information shared across individuals. Could the new template also capture more inter-individual differences, since we are moving towards the direction of precision medicine?

3. Did the authors release the transformations between standard templates and this new template? I think the transformations will be important if one wants to compare previous research findings in standard template spaces, to new findings in this new template space.

4. What are the demographics of the 1031 participants used to create this new template? What is the age range? Patients/healthy controls? Males/females? Ethnicities? Do you think these demographic

differences could also contribute to the differences in your evaluation measures?

5. The code for new template creation and optimization is not released.

6. Lines 140-142: "Note that oversampling a brain region does not provide extra information—the extra vertices are simply extra interpolations from the same acquired data." This sentence is not completely true. Let's say the MRI resolution is 1mm. If even the densely sampled area has an inter-vertex distance of >2mm (which is true based on Figure 1d), we cannot say extra vertices do not provide extra information.

7. Expressions like lines 121-122: "the variance across vertices for onavg was 8.41% compared to fsavg, and 7.97% compared to fslr" are not easy to understand. I would prefer to say: the variance/accuracy of xxx increases/decreases from xxx [onavg] to xxx [fslr]. Same with other places throughout the paper.

Author Rebuttal to Initial comments

Reviewers' Comments:

We thank both reviewers for their positive feedback and helpful comments. We have revised the manuscript based on the reviewers' suggestions, and we believe the manuscript is much stronger after the revision.

Reviewer #1:

Remarks to the Author:

This paper introduced a new surface template constructed from a large scale of MRI scans consisting of 1000+ subjects. The key difference between this template and other commonly used surfaces, such as fsaverage (FreeSurfer) or fs_LR (HCP), is the uniform sampling of the vertices on the anatomical surface manifold, rather than uniform sampling on the inflated spherical space.

The authors demonstrated the advantage of their surface template via

1) a multivariate pattern classification (MVPC) and 2) a representational similarity analysis (RSA), both on a naturalistic (movie) fMRI dataset (n=15). The results showed a consistent improvement of the accuracy in MVPC and intersubject correlation in RSA, compared to fsavg and fsLR. Furthermore, the authors also demonstrated that the computational time for the searchlight-based algorithms was reduced using the uniform template.

The idea of uniform sampling on the pial surface rather than the sphere is neat. The coarse-to-fine optimization procedure for the template construction is clear and straightforward. I do think there could be a huge potential in using this onavg template in a variety of neuroscientific tasks.

However, I'm not fully convinced by the demonstrations for the following reasons:

We thank the reviewer for the accurate summary and positive evaluation of our manuscript.

Please see below our point-to-point responses to the reviewer's comments.

1. The experiments should not be constrained by the specific experimental pipelines from authors' previous works. It will be much more convincing if the authors can show the onavg template is able to yield better performance in a wide spectrum of neuroscientific tasks, in order to claim onavg is a better template for "human neuroscience" as their title suggested as well as be suitable to the broad audience in this journal.

We thank the reviewer for the great suggestion. We used MVPA to demonstrate the advantages of the onavg template for two reasons. First, MVPA is widely used in neuroscientific data analysis. Second, MVPA relies on spatial patterns, which are directly affected by sampling density and uniformity. The advantages of the onavg template, in theory, generalizes to any neuroscientific data analysis which involves sampling density or spatial patterns on the cortical surface. We performed a series of new analyses to demonstrate these advantages in a wide range of settings, which we elaborate below.

We replicated the MVPA results with two independent datasets (*Raiders* and *Budapest*, in addition to the original *Forrest*; Supplementary Information Part 2, Figures S9, S10, S11, and S12), and added three new analyses to showcase the benefits of the onavg template in other commonly used neuroscientific methods. Specifically, using a substantial sample (Human Connectome Project (HCP), $n = 888$), we show that (a) local functional connectivity based on the onavg template has less bias from geometric distortions compared to other templates (Supplementary Information Part 4, Figure S14); (b) functional contrast maps, which are commonly used to localize functional areas, have better quality based on the onavg template

compared to other templates (Supplementary Information Part 4, Figure S15); and (c) individual differences in functional organization based on the onavg template are more prominent compared to other templates (Supplementary Information Part 4, Figure S16). These additional analyses demonstrate the broad application of the onavg template through three key topics of neuroscience—functional connectivity, functional localizers and contrasts, and individual differences. We added these new results to the Supplementary Information (Supplementary Information Part 4; Figures S14–S16).

2. Another downside of using the authors' current experiments is that the experimental pipeline seems too complicated. How could the authors prove it's the onavg template that results in improved performance, rather than some inherent bias of the experimental pipeline?

We thank the reviewer for pointing this out. For all our analyses comparing different surface templates, we carefully conducted them so that they were based on the same data, same preprocessing, same alignment method, same analysis code, same Python environment, and on the same node of the high-performance computing cluster. The only difference between the conditions (three different templates) is that the data were resampled to the specific surface space.

To further demonstrate the robustness of our results, in this revision, we repeated the MVPA analysis with two additional datasets and three different alignment methods, and we got similar results as in the original manuscript, further eliminating the possibility that the effects were specific to a dataset or an alignment method (Supplementary Information Part 2; Figures S9–S12).

We agree with the reviewer that certain steps are not necessary, given that the effects on MVPA results are consistent for different alignment methods. Therefore, we have removed the

hyperlignment step in the main text to simplify the pipeline, and put the hyperaligned results in Supplementary Information instead (Supplementary Information Part 1, Replication with alternative alignment methods, Figures S4 and S5). We believe this change has made the manuscript easier to follow and digest.

3. The sample size (in the demonstration experiments, not the template construction) is too small.

The reviewer is correct that smaller sample sizes are often associated with less statistical power, which might undermine the replicability of the results. In this study, we carefully designed the analysis, so that the variation in classification accuracy and RSA-ISC across conditions is only driven by resampling to different surface templates. By minimizing unwanted confounding variations, we observed large effect sizes of the differences, which were statistically significant even with a moderate sample size, which we will elaborate in the reply to Point #4 below.

It is still possible, however, that smaller sample sizes might violate certain statistical assumptions, especially those based on the central limit theorem. Therefore, to directly evaluate the replicability of our results, we repeated our analysis with two additional naturalistic viewing datasets (*Raiders*: $n = 23$; *Budapest*: $n = 21$), which were collected with different fMRI scanners and protocols, different movies, and different participants compared to the *Forrest* dataset used in the original analysis. We observed similar benefits of the onavg template on MVPA results in these two datasets, demonstrating the replicability of our results in a wide range of settings (Supplementary Figures S9–S12).

In addition, we used the larger HCP dataset ($n = 888$) and conducted a series of analyses of functional connectivity, contrast maps, and individual differences in cortical functional

architecture using data resampled into the onavg, fsavg, and fslr cortical models

(Supplementary Figures S14–S16), further resolving the concerns of limited statistical power with smaller sample sizes.

4. The usage of movie fMRI data is kind of tricky, as the brain response can be very complex. I don't see (and it's unlikely based on the boxplots) there will be statistical significance in MVPC accuracy nor in RSA-ISC between onavg result and fsavg/fslr result.

We thank the reviewer for pointing this out. The differences between onavg and fsavg/fslr were statistically significant for both MVPC accuracy and RSA-ISC. In the original manuscript we focused on the effects of increased data usage efficiency, which was a large effect (1/4 less data to achieve the same performance) and of great interest to the community. We have modified the manuscript to clarify the statistics:

The average accuracy across participants significantly increased from 13.3% (fsavg) and 13.2% (fslr) to 15.7% (onavg), both $t(14) > 10.0$, Cohen's $d > 2.60$, $P < 10^{-7}$. For all 15 out of 15 participants, the accuracy based on onavg was higher than based on other templates (Figure 2a).

The average RSA-ISC based on the onavg template was consistently higher than the average RSA-ISCs based on fsavg and fslr for all 15 participants, and the average RSA-ISC significantly increased from 0.094 (fsavg) and 0.094 (fslr) to 0.104 (onavg), both $t(14) > 28.2$, Cohen's $d > 7.30$, $P < 10^{-13}$ (Figure 2c).

The gray lines in the figure demonstrated that the accuracy and RSA-ISC were higher based on onavg than other templates for each individual participant, which corresponds to 15 consecutive successes in 15 binomial trials ($P = 3 \times 10^{-5}$, one-sided). Note that individual differences have a larger effect on accuracy and RSA-ISC than surface templates. To emphasize the effect of

surface templates, in the replication analyses we added two new supplementary figures on the group-average difference and its standard error, for classification accuracy and RSA-ISC, respectively:

Average difference in classification accuracy (Supplementary Figure S10):

Average difference in RSA-ISC (Supplementary Figure S12):

Together, these results show that classification accuracy and RSA-ISC based on the onavg template are higher than those based on other templates. These differences are statistically significant and are consistent across alignment methods and datasets.

5. There are many other domains that also rely on surface template besides naturalistic fMRI studies, such as anatomical/functional parcellations, cytoarchitecture studies linking ex vivo scans to in vivo MRI, vascular studies, etc.

With all being said above, I would suggest the authors conduct some very very basic experiments that people use every day across several different categories of neuroscientific tasks to demonstrate the benefit of using onavg. Just a few examples I can think of:

* Registration: register each subject from a large-scale dataset (e.g. HCP or UK Biobank, as long as it's independent of the OpenNeuro set) to the onavg template based on geometric features such as sulcal depth or curvature using any widely used registration tool e.g.

FreeSurfer, FSL, ANT, whatever. Once the pipeline is fixed, only change the registration target, i.e. swapping the onavg with fsavg and fslr, and compare the results. Do you see improved registration accuracy?

* resting-state fMRI: sample (perhaps the minimally preprocessed) rsfMRI data for each subject onto the three templates, simply compute connectivity (vertex-wise Pearson correlation matrix). Do you see a smaller variance in the connectivity across subjects?

* task fMRI: sample task data onto the three templates and just do a GLM regression against the block designs. Maybe just need one or a few simple tasks for the demonstration, e.g. finger tapping task or visual task where the responsive regions are well localized. Do you see a stronger task response averaged over subjects or do you see sharper boundaries between task and non-task regions?

The reviewer raised a very insightful question, that is, whether the group statistics of the 1,031 participants also affords a better *registration* target, in addition to a better *resampling* target demonstrated in the current manuscript. Note that the registration target and resampling target are usually the same for volume-based analysis (e.g., MNI152NLin2009cAsym), yet they are different for surface-based analysis. In surface-based analysis, the registration target is the group statistics of cortical folding patterns, such as sulcal depth and curvature, as the reviewer pointed out. The resampling target, however, is the location of a set of standard vertices on the cortical surface. For example, in this work, we registered all the participants using FreeSurfer's registration algorithm and registration target, and resampled data to different resampling targets (i.e., fsavg, fslr, and onavg).

We split the 1,031 participants into two groups while ensuring that participants from the same dataset were in the same group. We computed the average sulcal depth ("sulc") and curvature ("curv") maps for each group, and for each of the two measures, correlated the two average maps. The correlation coefficient was 0.998 and 0.994 for sulcal depth and curvature maps, respectively, which translates to split-half reliability of 0.999 and 0.997, respectively. In other words, the average maps based on the entire sample of 1,031 participants are very close to the group truth. In contrast, with 40 participants, the reliability was only 0.932 and 0.966, respectively, based on the Spearman-Brown prediction formula. Therefore, benefiting from the large sample size, the average maps based on the 1,031 participants have excellent quality and great potential to serve as better registration targets. We have made these average maps openly available as a DataLad dataset through <https://gin.g-node.org/neuroboros/core>.

Note that the quality of the registration also depends on the registration algorithm, the maps used as input, and the hyperparameter choices. For example, recently MSM (Robinson et al., 2014, 2018) was proposed as an alternative to FreeSurfer's registration algorithm (Fischl et al., 1999). MSMSulc takes only the average sulcal depth map as the input. In contrast, FreeSurfer also uses the average curvature map and other statistics, whereas MSMAll uses additional myelin and functional maps. The HCP Pipelines and the official MSM release use different regularization parameters for the MSM algorithm. An analysis of registration quality as a function of registration targets and these methodological choices may take a great amount of time and effort, and we think it is best to leave it to future research for a more comprehensive study of this topic.

We agree with the reviewer that it is critical to demonstrate the benefits of the onavg template in a wide range of scenarios. In addition to the classification and RSA analyses in the original manuscript, we performed three additional analyses on three key topics of neuroscience:

resting-state functional connectivity, functional contrast maps, and individual differences in cortical functional architecture. We have summarized the results of the analyses into a new part of the Supplementary Information:

Part 4: Functional connectivity, contrast maps, and individual differences

In the main text and the first two parts of this supplementary file, we demonstrated the advantages of the onavg template for various MVPA analyses and the robustness of these advantages across a wide range of conditions. These advantages are driven by the uniform sampling of the cerebral cortex, and in theory, apply to any analysis based on cortical surface.

In this part, we use the Human Connectome Project (HCP) dataset and showcase the advantages of the onavg template on three key topics of neuroscience: (a) resting-state functional connectivity, which is commonly used to study the intrinsic functional organization of the brain; (b) functional contrast maps, which is often used to localize functional regions of interest; and (c) individual differences in brain functional architecture, which is key to precision neuroscience and translational neuroscience.

For the first two analyses, we used the 3 T resting-state and task fMRI data of the HCP dataset, respectively. Both analyses were based on 888 participants who had complete 3 T fMRI data. The last analysis was based on the 7 T data with movie-watching, and it included 178 participants, which partly overlap with the 888 3 T participants (149 overlapped participants). One participant (221319) was excluded from the 7 T data analysis due to data quality (including the participant does not change any conclusions).

For conciseness, details of these analyses are not included in the reply. The results are summarized as Supplementary Figures S14–S16. In short, the onavg template reduces bias in

local functional connectivity, improves the quality of functional contrast maps, and affords better measures of individual differences in cortical functional architecture.

Some other minor comments:

- The authors claimed, "Note that oversampling a brain region does not provide extra information-the extra vertices are simply extra interpolations from the same acquired data". This seems to be a very bold claim and perhaps only applicable to the current scale of fMRI data. There are many other applications that need denser samples, e.g. studies of high res ex vivo data. That's why when people release data, surfaces are usually provided in different resolutions.

We thank the reviewer for pointing this out. We have taken it out of the manuscript to avoid potential confusions. The new sentence reads:

Note that undersampling a brain region permanently discards certain information, especially the information encoded in fine-grained spatial patterns.

We agree with the reviewer that it is critical to provide the template with different resolutions. We have prepared the onavg template in 7 different resolutions, from ico4 to ico128, to address various needs of different types of data. ico4, ico8, and ico16 can be used as sparse searchlights; ico32, ico48, and ico64 are typical resolutions of fMRI data; ico128 can be used for high-resolution structural data. Among these resolutions, ico32, ico64, and ico128 correspond to fsaverage5, fsaverage6, and fsaverage of FreeSurfer, respectively, which are the most commonly used resolutions.

The issue of uneven sampling exists for all traditional templates created using icosahedron subdivision, including all resolutions. The onavg template resolves the issue of uneven

sampling. For all the resolutions we tested, including ico32, ico64, and ico128, onavg affords much better uniformity in vertex spacing compared to traditional cortical surface templates (see also the reply to the point below).

- Results in ico64 (~2mm inter-vertex distance) were shown in supplementary materials. But this resolution is usually considered downsampled surfaces. Does onavg also do better than fsavg/fslr in higher resolution surfaces, e.g. 164k native mesh from FreeSurfer/HCP?

We agree with the reviewer that it is often necessary to use different resolutions of the template depending on the data type and analysis. For example, higher resolution is more often used with T1w data (~1 mm resolution) and morphometric analysis, compared to BOLD data (often acquired with 2–3 mm resolution) and functional analysis. Therefore, we released the onavg template with multiple spatial resolutions to accommodate different needs, including ico32, ico48, ico64, and ico128 (approximately 4 mm, 3 mm, 2 mm, and 1 mm resolution, respectively).

Different spatial resolutions of the onavg template were optimized using the same algorithm, and all of them demonstrated markedly higher spatial uniformity compared to other templates. Similar to the ico32 and ico64 resolutions (common resolutions for functional data) described in the main text, we found that the onavg template at the ico128 resolution (164k) also had significantly lower variations in inter-vertex distance (top row of the figure below) and vertex area (bottom row) throughout the cortex.

- It would be great to show, in the very first figure, how does the onavg template look like in both the pial and the spherical surface, and perhaps a zoomed-in version and compare that with fsavg/fslr, just settle an intuitive impression for the audience.

We thank the reviewer for this suggestion. After plotting the vertices on different surfaces, we realized that some differences are easier to notice by flipping the images of different templates back-and-forth than by placing them side-by-side. Therefore, we decided to create animations to highlight these differences, so that the readers can get an intuitive impression of the differences. Each animation is based on the data of a test participant, and it contrasts the distribution of cortical vertices between onavg and fsavg/fslr for both anatomical surfaces ("midthickness") and spherical surfaces on the test participant's brain. These animations, which cannot be inserted in this letter, can be found at https://feilong.github.io/tpl-onavg/vertex_properties/vertex_distribution.html

Albeit these comments, I find the study is interesting, worth pursuing, and can be a great contribution to the community if the authors can demonstrate its advantage over a range of most commonly used neuroscientific tasks.

We thank the reviewer again for the helpful comments and the positive evaluation of our work.

Reviewer #2:

Remarks to the Author:

Feilong et al. proposed a new surface template that solves the problem of the current standard surface templates: the vertices are unevenly distributed in anatomical space. With this new template, they showed that the classification accuracy of movie fMRI time points was higher, the inter-subject correlation of representational geometry was higher, and the computational time of searchlight-based algorithms was lower, compared to using standard surface templates. This paper is very well written and the authors demonstrated the applicability of this new template.

However, I have a few comments:

We thank the reviewer for the positive evaluation and the helpful comments. Please see below for the point-to-point responses to the questions.

1. All evaluation measures involved using hyperalignment. However, many studies do not use hyperalignment. For example, would the movie time points classification accuracy still increase?

The reviewer raised an important question. For all benchmarking analyses with naturalistic movie datasets, we repeated each analysis three times with different alignment methods—surface alignment (i.e., no hyperalignment), Procrustes hyperalignment, and warp hyperalignment. The classification accuracy and RSA-ISC based on the onavg template were higher than those based on traditional templates for all alignment methods. This was because for traditional templates, vertices were not evenly distributed on the cortical surface; this led to

loss of information during the resampling process, which affected all subsequent analyses, regardless of the alignment method. Therefore, the performance based on the onavg template was better for all three alignment methods.

As Reviewer #1 also pointed out, simplifying the analysis workflow will help readers digest the essence of the work, therefore, we decided to use the results based on surface alignment in the main text and move the hyperalignment results to Supplementary Information (Part 1, Replication with alternative alignment methods, Figures S4 and S5). The advantages of the onavg template using surface-aligned data were similar to using hyperaligned data.

2. Both the classification accuracy of movie time points, and the inter-subject correlation of representational geometry look for the common neuro-information shared across individuals. Could the new template also capture more inter-individual differences, since we are moving towards the direction of precision medicine?

We thank the reviewer for raising this important issue. We agree with the reviewer that individual differences in the brain are a key research topic in neuroscience, and understanding and modeling these differences will help develop individualized education and treatment, leading to better life outcomes.

To test whether onavg affords better measures of individual differences in the brain, we performed a new analysis using the 7 T fMRI data of the Human Connectome Project. The analysis was based on 178 participants who had movie-watching data. We built a functional template based on 89 training participants' data and used it to model the functional organization of the remaining 88 test participants (we excluded one participant for data quality). For each test participant, we split the movie data into two halves and computed an estimate of the participant's neural tuning based on each half of the data. The two estimated tuning matrices

from the same participant were more similar than those between different participants, and the prominence of individual differences can be measured using distinctiveness—normalized difference of within-participant similarity and between-participant similarity. For almost every participant, the distinctiveness based on the onavg template was higher than based on other templates (86 and 84 out of 88, respectively, for fsavg and fslr), indicating that individual differences are more prominent based on the onavg template. This was because the advanced sampling procedure of onavg better captures the information in brain response patterns, which includes both shared information and idiosyncratic information. Therefore, it improves both results that rely on shared information (e.g., ISC) and results that rely on idiosyncratic information (e.g., distinctiveness in neural tuning).

We have summarized the analysis and the results as Supplementary Information Part 4, Figure S16:

Figure S16. Amplified individual differences in cortical functional architecture. We performed the analysis using neural responses to the movie from 178 participants of the HCP 7 T dataset. We created a functional template using 89 participants' data and used the template to model the functional architecture of the remaining 88 participants (excluding 1 participant for data quality)³⁶. For each of the 88 test participants, we obtained two estimates of their neural tuning, one based on each half of the data. (a) The estimated tuning matrices of the same participant were much more similar than those from different participants. Based on these within-participant and between-participant similarities, we can estimate how distinctive each participant's neural tuning is. (b, c) For almost every participant, the distinctiveness based on the onavg template was higher than those based on fsavg and fslr (86 and 84 out of 88, respectively; both $t > 17.8$, $P < 10^{-30}$). (d) The average distinctiveness across participants based on the onavg template was consistently higher than those based on other templates across different amounts of data. Together, these results show that individual differences based on the onavg template are more prominent compared to those based on other templates.

3. Did the authors release the transformations between standard templates and this new template? I think the transformations will be important if one wants to compare previous research findings in standard template spaces, to new findings in this new template space.

We thank the reviewer for bringing up this important question, and we agree with the reviewer that making the transformations available will facilitate comparing scientific findings based on different template spaces, as well as encourage researchers to switch to the new template space.

We have prepared 182 pre-computed transformations between difference template spaces, including 42 transformations between different resolutions of the onavg template (ico4, ico8, ico16, ico32, ico48, ico64, and ico128), 98 transformations between onavg and other templates (fsavg: ico32, ico64, ico128; fslr: ico32, ico57, ico64, ico128), and 42 transformations between other templates. We have made these transformations openly available through the data

repository <https://gin.g-node.org/neuroboros/core>, with the following naming convention:

`{source}/mapping/to_{target}/{lr}h/on1031_trimmed/overlap-8div.npz`

To help researchers use these transformations, we have also prepared a tutorial with example code to resample different types of data with these transformations, openly available at:

https://feilong.github.io/tpl-onavg/how_to_use/space_travel.html

4. What are the demographics of the 1031 participants used to create this new template? What is the age range? Patients/healthy controls? Males/females? Ethnicities? Do you think these demographic differences could also contribute to the differences in your evaluation measures?

We thank the reviewer for the great question. One of the advantages of using OpenNeuro data is that the participants are aggregated through diverse studies, making them representative of the general population of neuroimaging study participants.

Among the 1031 participants, 471 are females, 408 are males, 2 are non-binary, and 150 are unknown. The mean age \pm standard deviation was 28.42 ± 14.57 yo, based on the 767 participants whose age information is available. The participants were mainly young adults, with a small portion of younger and older participants. The age range in years was 8–81, and the 10th and 90th percentiles were 18 and 43.76, respectively. Other demographic information, such as race, ethnicity, and handedness, are not available for most of the participants. For every participant, we checked the reconstructed cortical surface and excluded the participant if the reconstructed surface was problematic. The 1031 participants were the remaining participants after the exclusion, and their brains had no visible lesion or abnormality.

The current work aims to provide a single template that can be used for a wide range of participants. For certain populations, such as children whose gyrification is still under

development, a population-specific template might be beneficial if the template can be built from a similar amount of data from the specific population.

5. The code for new template creation and optimization is not released.

We thank the reviewer for the suggestion. We have added the code used for template creation and optimization to the code release for reproducibility purposes, which can be found at <https://feilong.github.io/tpl-onavg/optimization.html>

We have also updated the code release and added the code to replicate the MVPA analyses with two new datasets (*Raiders* and *Budapest*). The code includes whole-brain classification of movie time points, searchlight classification of movie segments, and searchlight analysis of RSA-ISC. The outline of the code can be found at <https://feilong.github.io/tpl-onavg/replication.html>

6. Lines 140-142: "Note that oversampling a brain region does not provide extra information—the extra vertices are simply extra interpolations from the same acquired data." This sentence is not completely true. Let's say the MRI resolution is 1mm. If even the densely sampled area has an inter-vertex distance of >2mm (which is true based on Figure 1d), we cannot say extra vertices do not provide extra information.

We thank the reviewer for pointing this out. The original sentence was supposed to refer to cases such as resampling data from 1mm to <1mm. We agree with the reviewer that the original statement can be misleading, and we have taken it out of the manuscript to avoid potential confusions. The new sentence reads:

Note that undersampling a brain region permanently discards certain information, especially the information encoded in fine-grained spatial patterns.

7. Expressions like lines 121-122: “the variance across vertices for onavg was 8.41% compared to fsavg, and 7.97% compared to fslr” are not easy to understand. I would prefer to say: the variance/accuracy of xxx increases/decreases from xxx [onavg] to xxx [fslr]. Same with other places throughout the paper.

We thank the reviewer for the suggestion. We have changed these expressions throughout the manuscript. Please see below a copy of the updated paragraphs:

[Variation in vertex properties across the cortex]

The anatomy-based sampling of the onavg template greatly reduced the heterogeneity of vertices in many ways. For inter-vertex distance, the variance decreased from 0.41 mm² (fsavg) and 0.43 mm² (fslr) to 0.03 mm² (onavg). For the cortical area occupied by each vertex, the variance decreased from 11.90 mm⁴ (fsavg) and 12.29 mm⁴ (fslr) to 0.55 mm⁴ (onavg). For the number of vertices in a 20 mm searchlight, the variance decreased from 1723.14 (fsavg) and 1858.20 (fslr) to 81.58 (onavg). For all these three vertex properties that we assessed, the variance across the cortex decreased dramatically from other templates to onavg (mean decrease = 94.23%; range: 91.59%–95.61%).

[Multivariate pattern classification of movie time points]

The average accuracy across participants significantly increased from 13.3% (fsavg) and 13.2% (fslr) to 15.7% (onavg), both $t(14) > 10.0$, $P < 10^{-7}$; for all 15 out of 15 participants, the accuracy based on onavg was higher than based on other templates (Figure 2a).

[Inter-subject correlation of representational geometry]

The average RSA-ISC based on the onavg template was consistently higher than the average RSA-ISCs based on fsavg and fslr for all 15 participants, and the average

RSA-ISC significantly increased from 0.094 (fsavg) and 0.094 (fslr) to 0.104 (onavg),
both $t(14) > 28.2$, $P < 10^{-13}$ (Figure 2c).

Decision Letter, first revision:

Dear Dr. Feilong,

Thank you for submitting your revised manuscript "A cortical surface template for human neuroscience" (NMETH-A52271A). It has now been seen by the original referees and their comments are below. The reviewers find that the paper has improved in revision, and therefore we'll be happy in principle to publish it in Nature Methods, pending minor revisions to satisfy the referees' final requests and to comply with our editorial and formatting guidelines.

TRANSPARENT PEER REVIEW

Please note: we allow redactions to authors' rebuttal and reviewer comments in the interest of confidentiality. If you are concerned about the release of confidential data, please let us know specifically what information you would like to have removed. Please note that we cannot incorporate redactions for any other reasons. Reviewer names will be published in the peer review files if the reviewer signed the comments to authors, or if reviewers explicitly agree to release their name. For more information, please refer to our FAQ page.

ORCID

Best regards,
Nina

Nina Vogt, PhD
Senior Editor
Nature Methods

Reviewer #1 (Remarks to the Author):

The authors well addressed the questions I had. I only have the following points mainly for discussion.

- The local functional connectivity analysis is a perfectly valid demonstration of the benefit of onavg over the other two atlases. However, the results for onavg (Fig. S14 c, g) look very spotty and noisy, isn't it? I'm wondering if there is anything off as the average result over 800+ subjects should be much smoother than that?

- Related to the first point, the local functional connectivity is not very commonly used, or at least not as common as the e.g. seeded correlation for whole brain functional connectivity analysis. It could be a very helpful add-on to show that using onavg template, the variance of the global connectivity (full correlation matrix or something equivalent) is lower than that using the other two atlases. But I also understand that the authors have already performed many experiments and the manuscript is lengthy, so defer to the authors.

- There is no obligations to do so and I totally understand the reason behind, but I still truly believe the manuscript might be better received by the larger neuroscience community if the author demonstrates the rsfMRI functional connectivity and task fMRI results in the main paper, perhaps even as the main result. This could be my bias of view, but for almost all of my colleagues, especially for those clinical scientists, they may not know any lab-specific method, but they do know what functional connectivity means and know how to interpret task localizer maps. Just my two cents.

Reviewer #2 (Remarks to the Author):

The authors have clearly responded to the questions I raised in the previous round of review. I only have one suggestion on their new analysis in reply to my second question. To investigate if their new surface template also has an advantage in capturing individual differences, they calculated each participant's "neural tuning". They showed that this "neural tuning" measure was more identifiable than fsaverage and fsLR templates. However, how "neural tuning" was calculated was not mentioned, and no reference was cited for the definition of this measure.

Reviewer #2 (Remarks on code availability):

The authors provided very detailed tutorials with the commands on the installation and usage of their new surface template, which I can run without a problem. They also provided the code to reproduce the results in their paper. However, due to the non-availability of the original dataset on my side, I did not run this part of code. The authors also warned in their tutorials in this part, "Do not create a new template for your own research unless you have strong reasons to do so. This will make it

unnecessarily difficult to compare your results with those of other studies and/or perform meta-analyses with your results."

Author Rebuttal, first revision:

We thank the reviewers again for the positive evaluation of our manuscript and the helpful comments. Please see below our point-to-point response to the remarks.

Reviewer #1:

Remarks to the Author:

The authors well addressed the questions I had. I only have the following points mainly for discussion.

- The local functional connectivity analysis is a perfectly valid demonstration of the benefit of onavg over the other two atlases. However, the results for onavg (Fig. S14 c, g) look very spotty and noisy, isn't it? I'm wondering if there is anything off as the average result over 800+ subjects should be much smoother than that?

The reviewer raised a very insightful question. We believe the spotty figure was caused jointly by (a) the nonlinear relationship between inter-vertex distance and the bias in local functional connectivity, and (b) individual differences in brain anatomy.

In the figure above, we systematically manipulated the distance between two vertices and assessed their functional connectivity using Pearson correlation. Similar to the original analysis, their time series were resampled from pure Gaussian noise. Note that the relationship between vertex distance and functional connectivity is a sigmoid curve rather than linear.

The onavg template was created so that the group-average inter-vertex distance has minimal variation throughout the cortex. However, even with the same group-average inter-vertex distance, the distance between a pair of vertices still varies across brains. Due to the nonlinear relationship between inter-vertex distance and local connectivity, the average local connectivity is not necessarily uniform throughout the brain.

- Related to the first point, the local functional connectivity is not very commonly used, or at least not as common as the e.g. seeded correlation for whole brain functional connectivity analysis. It could be a very helpful add-on to show that using onavg template, the variance of the global connectivity (full correlation matrix or something equivalent) is lower than that using the other two atlases. But I also understand that the authors have already performed many experiments and the manuscript is lengthy, so defer to the authors.

We agree with the reviewer that results based on whole brain functional connectivity will be a great add-on to our manuscript and a strong empirical evidence to illustrate the differences between cortical surface templates. We were tempted to perform this analysis during the previous round of revision, and we soon realized that the analysis was more computationally prohibitive than we had planned. In ico64 resolution, each vertex-by-vertex connectivity matrix takes about 50 GB of disk space, or 25 GB if we only use the vectorized upper triangle of this symmetric matrix. To compare the connectivity matrix across template spaces, we need to repeat this analysis three times, one for each surface template space (fsavg, fs1r, and onavg), which triples the required resources. We realized that this analysis requires more CPU hours and disk space than we originally planned for this work. To avoid further delays of the revision, we have decided to switch to the local connectivity analysis, and leave the whole brain functional connectivity analysis to future work.

- There is no obligations to do so and I totally understand the reason behind, but I still truly believe the manuscript might be better received by the larger neuroscience community if the author demonstrates the rsfMRI functional connectivity and task fMRI results in the main paper, perhaps even as the main result. This could be my bias of view, but for almost all of my colleagues, especially for those clinical scientists, they may not know any lab-specific method, but they do know what functional connectivity means and know how to interpret task localizer maps. Just my two cents.

We agree with the reviewer that the results based on functional connectivity and contrast maps are highly important, as they demonstrate the general applicability of the onavg template in neuroscientific research. We have moved the results from Supplementary Information into

Extended Data Figs. 7–9 to increase the accessibility of these results. We have also emphasized these results by enriching one related paragraph of the Results section. The paragraph now reads:

In the analyses above, we demonstrate the advantages of the onavg template using MVPA, which by definition relies on spatial patterns. These advantages, in theory, generalize to any neuroscientific data analysis which involves sampling density, uniformity, or spatial patterns on the cortical surface. To demonstrate the broad applicability of the onavg template, we used the Human Connectome Project (HCP) dataset to showcase the advantages of the onavg template on three key topics of neuroscience: (a) resting-state functional connectivity, which is commonly used to study the intrinsic functional organization of the brain (Extended Data Fig. 7); (b) functional contrast maps, which is often used to localize functional regions of interest (Extended Data Fig. 8); and (c) individual differences in brain functional architecture, which is key to precision neuroscience and translational neuroscience (Extended Data Fig. 9). Together, these results demonstrate that the onavg template affords various advantages for a wide range of neuroscientific studies, and these advantages are consistent across datasets and methodological choices.

Reviewer #2:

Remarks to the Author:

The authors have clearly responded to the questions I raised in the previous round of review. I only have one suggestion on their new analysis in reply to my second question. To investigate if their new surface template also has an advantage in capturing individual differences, they calculated each participant's "neural tuning". They showed that this "neural tuning" measure

was more identifiable than fsaverage and fsLR templates. However, how "neural tuning" was calculated was not mentioned, and no reference was cited for the definition of this measure. We thank the reviewer for pointing this out. We have added a new part in Supplementary Methods to enhance the clarity and replicability of this analysis (see below). We have also included the reference to our paper that defines the tuning matrix and its derivation in detail in both Supplementary Methods and the figure caption.

Individual differences in functional architecture

We used neural responses to the movie from 178 participants of the HCP 7 T dataset for this analysis. We used 89 participants' data (training data) to build a functional template and performed the analysis using the remaining 88 participants (test data; excluding 1 participant for data quality). Specifically, we modeled the neural responses to the movie of different participants as the same functional template transformed with individualized transforms. After applying the individualized transform to the template, we converted the transformed functional template into a tuning matrix, which depicts the neural tuning of the specific participant³⁶. The tuning matrix was designed so that (a) vertices (or individuals) that are more different in response time series are also more different in their tuning profiles (i.e., columns of the tuning matrix), (b) the same tuning matrix can be estimated from responses to different stimuli. This allowed us to compute two estimates of the tuning matrix for each participant by dividing the participant's movie data into two halves and estimating the tuning matrix from each half.

For each pair of participants, we computed the similarities between their tuning matrix estimates. The similarity was always based on tuning matrices estimated from different halves of the movie, that is, the first half of the movie for one participant, and the second half of the movie for the other. Therefore, for each pair of participants, we had two tuning matrix similarities, and we used the average of them as the between-participant similarity for this pair of participants. Similarly, for each participant, we

computed a within-participant similarity based on the two tuning matrix estimates of the participant.

After looping through all participants and participant pairs, for each participant, we had 1 within-participant similarity and 87 between-participant similarities. We computed the distinctiveness of the participant as the difference between the within-participant similarity and average between-participant similarity, divided by the standard deviation of between-participant similarity³⁶. In other words, the distinctiveness measures how far away the within-participant similarity was from the distribution of between-participant similarities, and higher distinctiveness means the differences between the participant and others are more prominent.

We repeated the analysis with different surface templates, and thus each participant had three distinctiveness values, one for each template. We compared the distinctiveness between onavg and the other two templates. We also repeated the analysis with different amounts of data, that is, 7 minutes, 14 minutes, ..., up to the entire movie data (60.91 minutes). The onavg template consistently demonstrated higher distinctiveness than other templates across different amounts of data (Extended Data Fig. 9).

Final Decision Letter:

Dear Feilong,

I am pleased to inform you that your Article, "A cortical surface template for human neuroscience", has now been accepted for publication in Nature Methods. The received and accepted dates will be April 12th, 2023 and June 6th, 2024. This note is intended to let you know what to expect from us over the next month or so, and to let you know where to address any further questions.

Over the next few weeks, your paper will be copyedited to ensure that it conforms to Nature Methods style. Once your paper is typeset, you will receive an email with a link to choose the appropriate publishing options for your paper and our Author Services team will be in touch regarding any additional information that may be required. It is extremely important that you let us know now whether you will be difficult to contact over the next month. If this is the case, we ask that you send us the contact information (email, phone and fax) of someone who will be able to check the proofs and deal with any last-minute problems.

After the grant of rights is completed, you will receive a link to your electronic proof via email with a

request to make any corrections within 48 hours. If, when you receive your proof, you cannot meet this deadline, please inform us at rjsproduction@springernature.com immediately.

Please note that *Nature Methods* is a Transformative Journal (TJ). Authors may publish their research with us through the traditional subscription access route or make their paper immediately open access through payment of an article-processing charge (APC). Authors will not be required to make a final decision about access to their article until it has been accepted. Find out more about Transformative Journals

If you are active on Twitter/X, please e-mail me your and your coauthors' handles so that we may tag you when the paper is published.

Please note that you and any of your coauthors will be able to order reprints and single copies of the issue containing your article through Nature Portfolio's reprint website, which is located at

<http://www.nature.com/reprints/author-reprints.html>. If there are any questions about reprints please send an email to author-reprints@nature.com and someone will assist you.

Best regards,
Nina

Nina Vogt, PhD
Senior Editor
Nature Methods